# Domain-invariant Feature Exploration for Domain Generalization

**Wang Lu** *luwang@ict.ac.cn*
*Institute of Computing Technology, Chinese Academy of Sciences*

**Jindong Wang**[*] *jindong.wang@microsoft.com*
*Microsoft Research Asia*

**Haoliang Li** *haoliang.li@cityu.edu.hk*
*City University of Hong Kong*

**Yiqiang Chen** *yqchen@ict.ac.cn*
*Institute of Computing Technology, Chinese Academy of Sciences. Peng Cheng Laboratory*

**Xing Xie** *xingx@microsoft.com*
*Microsoft Research Asia*

**Reviewed on OpenReview:** *https://openreview.net/forum?id=0xENE7HiYm*

## Abstract

Deep learning has achieved great success in the past few years. However, the performance of deep learning is likely to impede in face of non-IID situations. Domain generalization (DG) enables a model to generalize to an unseen test distribution, i.e., to learn domain-invariant representations. In this paper, we argue that domain-invariant features should be originating from both internal and mutual sides. Internal invariance means that the features can be learned with a single domain and the features capture intrinsic semantics of data, i.e., the property within a domain, which is agnostic to other domains. Mutual invariance means that the features can be learned with multiple domains (cross-domain) and the features contain common information, i.e., the transferable features w.r.t. other domains. We then propose DIFEX for Domain-Invariant Feature EXploration. DIFEX employs a knowledge distillation framework to capture the high-level Fourier phase as the internally-invariant features and learn cross-domain correlation alignment as the mutually-invariant features. We further design an exploration loss to increase the feature diversity for better generalization. Extensive experiments on both time-series and visual benchmarks demonstrate that the proposed DIFEX achieves state-of-the-art performance.

## 1 Introduction

Over the past years, machine learning, especially deep learning has achieved remarkable success across wide application areas such as computer vision (He et al., 2016) and natural language processing (Vaswani et al., 2017). However, machine learning generally assumes that the training and test datasets are identically and independently distributed (IID) (Esfandiari et al., 2021; Xu et al., 2019), which may not hold in reality. Such non-IID issue is more practical and challenging. For instance, we expect that a model that recognizes the activities of a child can generalize well on the data from an adult, even if their data distributions are different due to different lifestyles and body shapes.

When the target data is available for training (e.g., the adult data is accessible), domain adaptation (DA) (Wilson & Cook, 2020) can be employed to handle the non-IID issue and learn an adapted model

---

[*]Corresponding author. Code: `https://github.com/jindongwang/transferlearning/tree/master/code/DeepDG`.

for the target domain. However, a more practical situation is when the target domain is *unseen* in training, which makes existing DA approaches not feasible. Domain generalization (DG), or out-of-distribution generalization is one of the most popular research topics that aims to solve such problems (Wang et al., 2022). DG learns a generalized model from multiple training datasets that can generalize well on an unseen dataset. There are several approaches for DG, such as data augmentation (Tobin et al., 2017; Shankar et al., 2018; Lu et al., 2022) that increase data diversity and meta-learning (Li et al., 2018a; Balaji et al., 2018) that learns generally transferable knowledge by simulating multiple tasks.

Different from these two approaches, domain-invariant feature learning is a popular DG strategy that aims to learn representations that remain invariant across different domains, thus benefiting cross-domain generalization. For instance, Ganin et al. (2016) designed the domain-adversarial neural network (DANN) using adversarial training, where they tried to confuse the domain classifier such that it could not distinguish which domains the features belonged to, thus achieving domain-invariant learning. Similar to DANN, many other methods are proposed to learn domain-invariant features for DG and achieved great success (Muandet et al., 2013; Li et al., 2018b; Matsuura & Harada, 2020; Sun & Saenko, 2016).

The popularity and effectiveness of domain-invariant learning naturally motivate us to seek the rationale behind this kind of approach: *what* are the domain-invariant features and *how* to further improve its performance on DG? Previously, Zhao et al. (2019) showed that feature alignments across domains are not enough for domain adaptation and they paid attention to label functions. However, due to unseen targets in DG, label functions cannot be accessed and Bui et al. (2021) utilized meta-Domain Specific-Domain Invariant to solve the above problem. In this paper, we take a deep analysis of the domain-invariant features. Specifically, we argue that domain-invariant features should be originating from both *internal* and *mutual* sides: the internally-invariant features capture the intrinsic semantic knowledge of the data while the mutually-invariant features learn the cross-domain transferable knowledge. On the one hand, the internally-invariant features are born with the input data that do not change with the existence of other domains. On the other hand, the mutually-invariant features focus on harnessing the cross-domain transferable knowledge that is mined from different distributions. Therefore, the integration of these features can ensure better generalization to unseen domains.

We propose *DIFEX* for D̲omain-I̲nvariant F̲eature EX̲ploration of the internally- and mutually-invariant representations. Specifically, for internal features, DIFEX employs a knowledge distillation framework to capture the high-level semantics using Fourier transform. For mutual features, DIFEX utilizes correlation alignment to align the feature distributions of any two domains. To allow mutual feature exploration, DIFEX further adds a regularization to maximize their divergence. We conduct extensive experiments across both image data and time series datasets to comprehensively show the advantage of DIFEX. Results show that our DIFEX outperforms other recent baselines in all datasets. Our contributions are mainly three-fold:

1. We propose DIFEX for Domain-Invariant Feature EXploration of both internally- and mutually-invariant representations. We further develop a regularization to maximize their divergence to allow for more feature exploration.[1]

2. Comprehensive experiments on both visual and time series in several different settings demonstrate the superiority and universality of DIFEX.

3. Our method can inspire the community to explore more diverse features from both inter- and intra-domain simultaneously, which may be a promising direction for future DG research.

## 2 Related Work

### 2.1 Domain Generalization

Existing domain generalization approaches (Wang et al., 2022) mainly consist of data augmentation (Shankar et al., 2018; Tobin et al., 2017), domain-invariant feature learning (Muandet et al., 2013; Ganin et al., 2016;

---

[1]To the best of our knowledge, DIFEX is the *first* work that proposes these two concepts (internally- and mutually-invariant representations), and utilizes inheritance and exploration simultaneously in DG.

Li et al., 2018b), and meta-learning (Li et al., 2018a; Balaji et al., 2018) techniques. Most of existing work pays attention to some specific applications such as computer vision and reinforcement learning. Our work belongs to domain-invariant feature learning-based DG, where we mainly focus on common domain-invariant feature learning methods which can be directly applied to different tasks.

## 2.2 Domain-invariant Representation Learning

Domain-invariant representation learning (Johansson et al., 2019; Ben-David et al., 2010) has long been a popular solution to addressing domain shift problems such as domain adaptation (Ganin & Lempitsky, 2015; Ganin et al., 2016; Zhao et al., 2019) and domain generalization (Li et al., 2018b; Muandet et al., 2013; Matsuura & Harada, 2020). Specifically for DG, learning domain-invariant representations is critical since DG focuses on the invariance across domains. Such domain invariance is achieved can be achieved by explicit feature alignment (Li et al., 2018b; Motiian et al., 2017), adversarial learning (Ganin et al., 2016; Li et al., 2018c; Rahman et al., 2020), or disentanglement (Liu et al., 2021; Piratla et al., 2020; Ilse et al., 2020). Recently, some work pointed that only simple alignments maybe not enough for generalization (Bui et al., 2021). Our work shares the same goal with them, but with more possibilities to explore other invariant features that allow more diversities to increase the generalization ability. Our DIFEX may seem similar to the disentanglement framework from many existing efforts since it also learns two kinds of features and then combines them for final prediction. However, it significantly differs from them since we are actually learning two kinds of domain-invariant features, while disentanglement often learns invariant and specific features (Zhao et al., 2019).

## 2.3 Fourier Features

Recently, some work paid attention to Fourier phase in visual recognition (Yang & Soatto, 2020; Xu et al., 2021). FDA (Yang & Soatto, 2020) introduced the Fourier perspective into domain adaptation. It could generate target-like images for training by simply replacing a small area in the centralized amplitude spectrum of a source image with that of a target image. FACT (Xu et al., 2021) illustrated that Fourier phase information contained high-level semantics and was not easily affected by domain shifts. It utilized the mixup (Zhang et al., 2018) technique to enlarge the diversity of Fourier amplitude without changing labels and thereby learned a model that was insensitive to Fourier amplitude. Another work (Xingchen et al., 2021) utilized Fourier amplitude (style) to calibrate the target domain style on the fly. However, it required some operations when testing. Some other work tried to control Fourier amplitude during training for better generalization (Lin et al., 2022; Zheng et al., 2022). These work are designed for the visual field, although much work (Oppenheim et al., 1979; Oppenheim & Lim, 1981; Hansen & Hess, 2007) have demonstrated that the phase component retains most of the high-level semantics in original signals, which suggests that Fourier phase may be a kind of universal domain-invariant features.

# 3 Methodology

## 3.1 Problem Formulation

We follow the problem definition in (Wang et al., 2022) to formulate domain generalization in a $C$-class classification setting. In domain generalization, multiple labeled source domains, $\mathcal{S} = \{\mathcal{S}^i | i = 1, \cdots, M\}$ are given, where $M$ is the number of sources. $\mathcal{S}^i = \{(\mathbf{x}_j^i, y_j^i)\}_{j=1}^{n_i}$ denotes the $i^{th}$ domain, where $n_i$ denotes the number of instances in $\mathcal{S}^i$. The joint distributions between each pair of domains are different: $P_{XY}^i \neq P_{XY}^j, 1 \leq i \neq j \leq M$. The goal of domain generalization is to learn a robust and generalized predictive function $h : \mathcal{X} \rightarrow \mathcal{Y}$ from the $M$ training sources to achieve minimum prediction error on an unseen target domain $\mathcal{S}_{test}$ with unknown joint distribution, i.e., $\min_h \mathbb{E}_{(\mathbf{x},y) \in \mathcal{S}_{test}}[\ell(h(\mathbf{x}), y)]$, where $\mathbb{E}$ is the expectation and $\ell(\cdot, \cdot)$ is the loss function. All domains, including source domains and unseen target domains, have the same input and output spaces, i.e., $\mathcal{X}^1 = \cdots = \mathcal{X}^M = \mathcal{X}^T \in \mathbb{R}^m$, where $\mathcal{X}$ is the $m$-dimensional instance space, and $\mathcal{Y}^1 = \cdots = \mathcal{Y}^M = \mathcal{Y}^T = \{1, 2, \cdots, C\}$, where $\mathcal{Y}$ is the label space.

### 3.2 Motivation

Existing literature (Yang et al., 2020; Yang & Soatto, 2020; Xu et al., 2021) show that Fourier phase information contains high-level semantics that are not easily affected by domain shifts in the visual field. Many early studies (Oppenheim et al., 1979; Oppenheim & Lim, 1981; Hansen & Hess, 2007) also conclude that in the Fourier spectrum of signals, the phase component retains most of the high-level semantics in the original signals, while the amplitude component mainly contains low-level statistics. Thus, Fourier phase features can act as the internally-invariant features, which, of course, are domain-invariant.

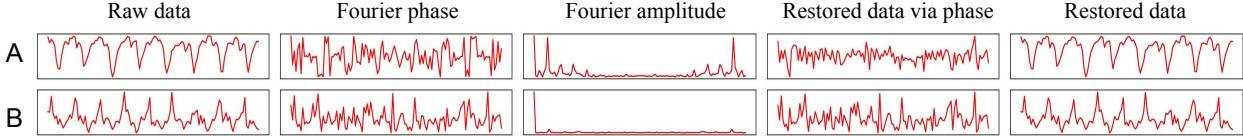

Figure 1: Fourier features as internally-invariant representations.

Is Fourier phase sufficient for DG? Figure 1 shows sensor readings of the walking activity collected from two persons in the UCI DSADS dataset (Barshan & Yüksek, 2014).[2] We apply Fourier transform to these two samples and compute their corresponding phase and amplitude. From the results, it is obvious that Fourier phases are similar between two different persons while there exist large gaps in Fourier amplitude. Therefore, Fourier phase information can perfectly act as domain-invariant features to help generalization. We clearly see that the raw data of *A* and *B* contains similar periodicity information that could be further utilized for generalization. However, when we restore data with Fourier phases and the sample amplitude, the two restored samples both lose some information compared with raw data such as the periodicity of walking. Thus, Fourier phase along would fail to capture the commonness from multiple training domains since they only focus on their own spectrum without considering the multi-domain statistics. Moreover, some existing work pointed that only simple alignments maybe not enough for generalization (Bui et al., 2021).

### 3.3 DIFEX

We propose Domain-Invariant Feature EXploration to learn both internally- and mutually-invariant features for domain generalization, short as DIFEX. As shown in Figure 2, DIFEX takes inputs from multiple training domains. Then, after a common feature extractor, we enforce the network to learn the internally- and mutually-invariant features. Subsequently, these features are concatenated to form the invariant features, which can then be used for classification. Furthermore, to allow the network to explore more diversity, we propose the exploration loss to regularize these features by maximizing their divergence.

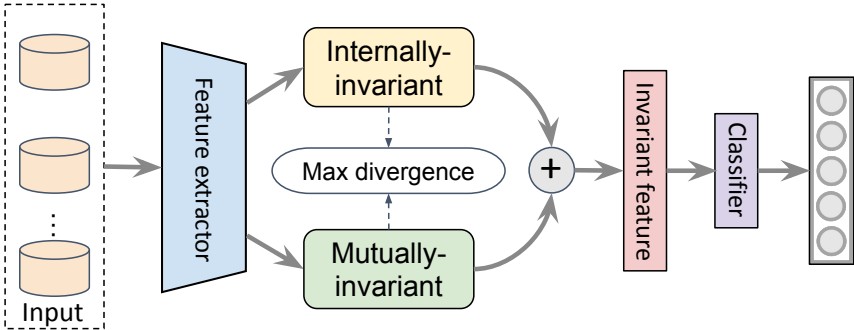

Figure 2: The framework of DIFEX.

---

[2]We do not perform the shift operation, but the schematics have demonstrated that amplitude contains very little information.

### 3.3.1 A distillation framework to learn internally-invariant features

We show how to obtain internally-invariant features with Fourier phase information. As shown in Figure 3, we utilize a distillation framework to obtain Fourier phase information for classification with raw data. Knowledge distillation is a simple framework to encourage different networks to contain particular characteristics (Hinton et al., 2015; Romero et al., 2014).

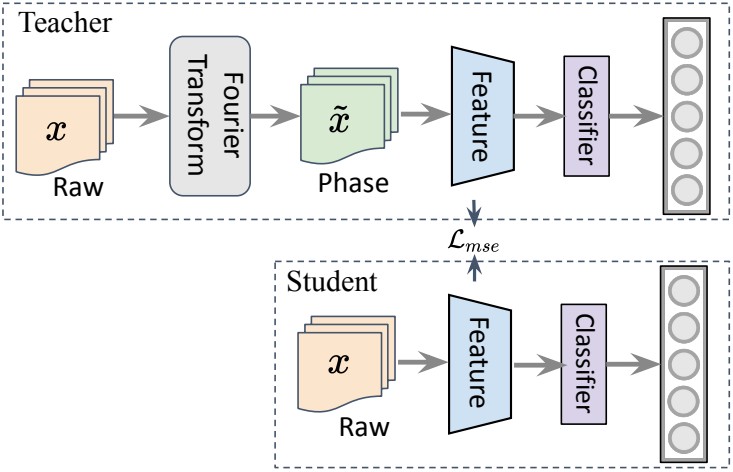

Figure 3: Distillation framework to learn internally-invariant features using Fourier transform.

Concretely speaking, the teacher network utilizes Fourier phase information and class labels as inputs and outputs, respectively, to obtain Fourier phase information features for classification. According to (Xu et al., 2021), the Fourier transformation $\mathcal{F}(\mathbf{x})$ for a single-channel two-dimensional data $\mathbf{x}$ is formulated as:

$$\mathcal{F}(\mathbf{x})(u,v) = \sum_{h=1}^{H-1}\sum_{w=0}^{W-1} \mathbf{x}(h,w)e^{-j2\pi\left(\frac{h}{H}u+\frac{w}{W}v\right)}, \tag{1}$$

where $u$ and $v$ are indices. $H$ and $W$ are the height and the width, respectively. Fourier transformation can be calculated with the FFT algorithm (Nussbaumer, 1981) efficiently. The phase component is then expressed as:

$$\mathcal{P}(x)(u,v) = \arctan\left[\frac{I(x)(u,v)}{R(x)(u,v)}\right], \tag{2}$$

where $R(x)$ and $I(x)$ represent the real and imaginary parts of $\mathcal{F}(x)$, respectively. For data with several channels, the Fourier transformation for each channel is computed independently to obtain the corresponding phase information. We denote Fourier phase of $\mathbf{x}$ as $\tilde{\mathbf{x}}$, then, the teacher network is trained using $(\tilde{\mathbf{x}}, y)$:

$$\min_{\theta_T^f,\theta_T^c} \mathbb{E}_{(\mathbf{x},y)\sim P^{tr}}\mathcal{L}_{cls}(G_T^c(G_T^f(\tilde{\mathbf{x}})), y), \tag{3}$$

where $\theta_T^f$ and $\theta_T^c$ are the learnable parameters of feature extractor $(G_T^f)$ and classification layer $(G_T^c)$ of the teacher network. $P^{tr}$ is distribution of training data, $\mathbb{E}$ denotes expectation, and $\mathcal{L}_{cls}$ is cross-entropy loss which is a common function for classification.

Once obtaining the teacher network, $G_T$, we use feature knowledge distillation to guide the student network to learn the Fourier information. Such a distillation is formulated as:

$$\min_{\theta_S^f,\theta_S^c} \mathbb{E}_{(\mathbf{x},y)\sim P^{tr}}\ell_c(G_S^c(G_S^f(\mathbf{x})), y) + \lambda_1\mathcal{L}_{mse}(G_S^f(\mathbf{x}), G_T^f(\tilde{\mathbf{x}})), \tag{4}$$

where $\theta_S^f$ and $\theta_S^c$ are the learnable parameters of feature extractor $(G_S^f)$ and classification layer $(G_S^c)$ of the student network. $\lambda_1$ is a tradeoff hyperparameter and $\mathcal{L}_{mse}$ is the MSE loss which can make features of the student network close to features of the teacher network.

### 3.3.2 Explore mutually-invariant features

As discussed early, Fourier phase features alone are insufficient to obtain enough discriminative features for classification. Thus, we explore the mutually-invariant features by leveraging the cross-domain knowledge contained in multiple training domains.[3] Specifically, given two domains $\mathcal{S}^i, \mathcal{S}^j$, we align their second-order statistics (correlations) using the correlation alignment approach (Sun & Saenko, 2016):

$$\mathcal{L}_{align} = \frac{2}{N \times (N-1)} \sum_{i \neq j}^{N} ||\mathbf{C}^i - \mathbf{C}^j||_F^2, \tag{5}$$

where $\mathbf{C}^i = \frac{1}{n_i-1}(\mathbf{X}^{i^T}\mathbf{X}^i - \frac{1}{n_i}(\mathbf{1}^T\mathbf{X}^i)^T(\mathbf{1}^T\mathbf{X}^i))$ is the covariance matrix and $||\cdot||_F$ denotes the matrix Frobenius norm.

Since there may exist duplication and redundancy between the internally-invariant features and mutually-invariant features, we expect that two parts can extract different invariant features as much as possible. This allows more diversities in features that are beneficial to generalization. Towards this goal, we regularize the distance between the internally-invariant ($\mathbf{z}_1$) and mutually-invariant ($\mathbf{z}_1$) features by maximizing their divergence, which we call the *exploration loss*:

$$\mathcal{L}_{exp}(\mathbf{z}_1, \mathbf{z}_2) = -d(\mathbf{z}_1, \mathbf{z}_2), \tag{6}$$

where $d(\cdot, \cdot)$ is a distance function and we simply utilize the $L2$ distance for simplicity: $\mathcal{L}_{exp} = -||\mathbf{z}_1 - \mathbf{z}_2||_2^2$.

### 3.3.3 Summary of DIFEX

To sum up, our method is split into two steps. First, we optimize Eq. 1. Second, we optimize the following objective:

$$\min_{\theta_f, \theta_c} \mathbb{E}_{(\mathbf{x},y) \sim P^{tr}} \mathcal{L}_{cls}(G_c(G_f(\mathbf{x})), y) + \lambda_1 \mathcal{L}_{mse}(\mathbf{z}_1, G_T^f(\tilde{\mathbf{x}})) + \lambda_2 \mathcal{L}_{align} + \lambda_3 \mathcal{L}_{exp}(\mathbf{z}_1, \mathbf{z}_2), \tag{7}$$

where $G_c$ and $G_f$ are the classification layer and the feature net respectively while $\theta_c$ and $\theta_f$ are corresponding parameters. $\lambda_1$, $\lambda_2$, and $\lambda_3$ are hyperparameters. Eq. 7 contains four objectives: classification, internally-invariant feature learning, mutually-invariant feature learning, and exploration of diverse features. The first and the third terms are two common objectives in invariant representation learning for domain generalization, and they are not enough according to the existing work Zhao et al. (2019); Bui et al. (2021). With the help of exploration of internally-invariant features and diverse features (the second and the last terms), we can alleviate the above problems to achieve better performance, which is proved in later experiments. For the current implementation, we mainly tune the hyperparameters, which balances these four terms. In the future, we plan to design more heuristic ways to automatically determine the hyperparameters.

As for inference, when a data sample $\mathbf{x}$ comes, we can predict its label by a simple forward-pass:

$$y = \arg\min G_c(G_f(\mathbf{x})). \tag{8}$$

### 3.4 Discussions

An alternative to learning Fourier features is to directly use the teacher network, and then use another feature extractor to extract mutually-invariant features. However, this would seriously increase the model size for inference as it requires two feature extractors. In addition, it also requires performing FFT for each sample as another input which is time-consuming for inference. Although our distillation framework would introduce more parameters in the training phase, it will not increase the model size and additional inputs for inference, which we believe is more useful in a real deployment.

One may argue that DIFEX may fail if there is only one training domain thus no mutually-invariant features. We provide a positive answer to this situation with virtual domain labels, i.e., domain alignment training on one domain by giving random domain labels. This simple trick seems effective in real experiments.

---

[3]Mutually-invariant features focus on harnessing the cross-domain transferable knowledge that is mined from different distributions. We think alignments among sources can bring common knowledge of sources.

## 4 Experiments

We extensively evaluate our method in both visual and time-series domains including image classification, sensor-based human activity recognition, EMG recognition, and single-domain generalization. The training data are randomly split into two parts: 80% for training and 20% for validation. The best model on the validation split is selected to evaluate the target domain. For fairness, we re-implement seven recent strong comparison methods following DomainBed (Gulrajani & Lopez-Paz, 2021): ERM, DANN (Ganin et al., 2016), CORAL (Sun & Saenko, 2016), Mixup (Zhang et al., 2018), GroupDRO (Sagawa et al., 2020), RSC (Huang et al., 2020), ANDMask (Parascandolo et al., 2021), FACT (Xu et al., 2021) and SWAD (Cha et al., 2021). In addition, for specific applications, we also add some latest comparison methods to compare. Please note that although our method contains two parts of features, we maintain the same architecture to predict for fairness, which means that both the dimension of internally-invariant features and the dimension of the mutually-invariant features are only half of the dimension of the other methods.

### 4.1 Image classification

Table 1: Accuracy on Digits-DG. The **bold** and underline items are the best and the second-best results. Results in the parentheses are the performance reported in other papers.

| Source | Target | ERM | DANN | CORAL | Mixup | GroupDRO | RSC | ANDMask | SWAD | DIFEX |
|---|---|---|---|---|---|---|---|---|---|---|
| MM,SV,SY | M | 97.55(95.80) | 97.77 | 97.62 | 97.50(94.20) | 97.48 | 97.78 | 96.85 | 97.37 | **97.82** |
| M,SV,SY | MM | 55.52(58.80) | 55.62 | 57.68 | **57.95**(56.50) | 53.47 | 56.27 | 56.00 | 55.77 | 57.90 |
| M,MM,SY | SV | 59.98(61.70) | 61.85 | 57.82 | 54.75(63.30) | 55.63 | 62.38 | 59.47 | 59.00 | **64.30** |
| M,MM,SV | SY | 89.25(78.60) | 89.37 | 90.12 | 89.80(76.70) | **92.15** | 89.25 | 88.17 | 87.92 | 89.98 |
| AVG | - | 75.58(73.70) | 76.15 | 75.81 | 75.00(72.70) | 74.68 | 76.42 | 75.12 | 75.02 | **77.50** |

Table 2: Accuracy on PACS. The **bold** and underline items are the best and the second-best results.

| Source | Target | ERM | DANN | CORAL | Mixup | GroupDRO | RSC | ANDMask | SWAD | DIFEX |
|---|---|---|---|---|---|---|---|---|---|---|
| C,P,S | A | 77.00(77.63) | 78.71 | 77.78 | 79.10(76.80) | 76.03 | 79.74(83.43) | 76.22 | **81.93** | 80.86 |
| A,P,S | C | 74.53(76.77) | 75.30 | 77.05 | 73.46(74.90) | 76.07 | 76.11(80.31) | 73.81 | 76.45 | **77.60** |
| A,C,S | P | 95.51(95.85) | 94.01 | 92.63 | 94.49(95.80) | 91.20 | **95.57**(95.99) | 91.56 | 94.67 | 95.57 |
| A,C,P | S | 77.86(69.50) | 77.83 | **80.55** | 76.71(66.60) | 79.05 | 76.64(80.85) | 78.06 | 76.79 | 79.49 |
| AVG | - | 81.22(79.94) | 81.46 | 82.00 | 80.94(78.50) | 80.59 | 82.01(85.15) | 79.91 | 82.46 | **83.38** |

Table 3: Accuracy on VLCS. The **bold** and underline items are the best and the second-best results.

| Source | Target | ERM | DANN | CORAL | Mixup | GroupDRO | RSC | ANDMask | SWAD | DIFEX |
|---|---|---|---|---|---|---|---|---|---|---|
| L,S,V | C | 93.64 | 94.49 | 96.33 | 96.18 | **97.81** | 96.89 | 93.50 | 94.56 | 96.61 |
| C,S,V | L | 60.05 | 64.34 | 64.42 | 63.55 | 62.24 | 62.91 | 64.83 | 61.78 | **67.21** |
| C,L,V | S | 68.46 | 67.15 | 68.65 | 68.86 | 69.23 | 69.07 | 63.28 | 67.82 | **74.31** |
| C,L,S | V | 74.02 | 72.69 | 70.73 | 71.95 | 70.73 | 70.38 | 68.87 | 67.89 | **75.24** |
| AVG | - | 74.04 | 74.67 | 75.03 | 75.13 | 75.00 | 74.81 | 72.62 | 73.01 | **78.34** |

**Datasets and implementation details** We adopt three conventional DG benchmarks. (1) **Digits-DG** (Zhou et al., 2020) contains four digit datasets: MNIST (LeCun et al., 1998), MNIST-M (Ganin & Lempitsky, 2015), SVHN (Netzer et al., 2011), SYN (LeCun et al., 1998). The four datasets differ in font style, background, and image quality. Following (Zhou et al., 2020), we utilize their selected 600 images per class per dataset. (2) **PACS** (Li et al., 2017) is an object classification benchmark with four domains (photo, art-painting, cartoon, sketch). There exist large discrepancies in image styles among different domains. Each domain contains seven classes and there are 9,991 images in total. (3) **VLCS** Fang et al. (2013) comprises photographic domains (Caltech101, LabelMe, SUN09, VOC2007). It contains 10,729 examples of 5 classes. For each dataset, we simply leave one domain as the test domain which is unseen in training while the others as the training domains.

For the architecture, we use ResNet-18 for PACS and VLCS, and Digits-DG dataset uses DTN as the backbone following Liang et al. (2020). Since splits, learning epochs, and some other factors all have influences on results, we mainly compare with the methods extended by ourselves for fairness. The maximum training epoch is set to 120. The initial learning rate for Digits-DG is 0.01 while 0.005 for the other two datasets. The learning rate is decayed by 0.1 twice at the 70% and 90% of the max epoch respectively.

**Results**  The results on three image benchmarks are shown in Table 1, 2, and 3, respectively. [4] Overall, our method has the average improvements of 1.08%, 0.92%, and 3% average accuracy than the second best method on three datasets. Visual classification for domain generalization is a challenging task, and it is difficult to have an improvement over 1%. As can be seen in these tables, the second-best method only has a slight improvement compared to the third one. This demonstrates the great performance of our approach in these datasets. Moreover, we see that alignments across domains and exploiting characteristics of data own can both bring remarkable improvements. In addition, some latest methods such as ANDMask even perform worse than ERM on some benchmarks, indicating that generalizing to unseen domains is really challenging.

### 4.2 Sensor-based human activity recognition

**Datasets and implementation details**  We also evaluate our method on a cross-dataset DG benchmark with four different datasets on human activity recognition (HAR). The four datasets are: UCI daily and sports dataset (**DSADS**) (Barshan & Yüksek, 2014) consists of 19 activities collected from 8 subjects wearing body-worn sensors on 5 body parts. And it contains about 1140000 samples with three sensors. **USC-HAD** (Zhang & Sawchuk, 2012). USC-SIPI human activity dataset (USC-HAD) composes of 14 subjects (7 male, 7 female, aged from 21 to 49) executing 12 activities with a sensor tied on the front right hip. And it contains 5441000 samples with two sensors. **UCI-HAR** (Anguita et al., 2012). UCI human activity recognition using smartphones data set (UCI-HAR) is collected by 30 subjects performing 6 daily living activities with a waist-mounted smartphone. And it contains 1310000 samples with two sensors. **PAMAP2** (Reiss & Stricker, 2012). PAMAP2 physical activity monitoring dataset (PAMAP2) contains data of 18 different physical activities, performed by 9 subjects wearing 3 sensors. And it has 3850505 samples with three sensors.

To evaluate our method thoroughly for time-series data, we design three settings: Cross-Person, Cross-Position, and Cross-Dataset generalization with these four datasets.:

- **Cross-Person Generalzation.**  We split DSADS, USC-HAD, and PAMAP [5] For each dataset, we split data into four parts according to the persons and utilize 0,1,2, and 3 to denote four domains. For example, 14 persons in USC-HAD are divided into four groups, $[1, 11, 2, 0], [6, 3, 9, 5], [7, 13, 8, 10], [4, 12]$ where different numbers denote different persons. We try our best to make each domain have a similar number of data.

- **Cross-Position Generalization.** We choose DSADS since it contains sensors worn on five different positions. Therefore, we split DSADS into five domains according to sensor positions.

- **Cross-Dataset Generalization.** To perform Cross-dataset generalization for HAR, we need to unify inputs and labels of each dataset first. Each dataset corresponds to a different domain. Six common classes are selected. Two sensors from each dataset that belong to the same position are selected and data is down-sampled to ensure the dimension of data same.

The HAR model contains two blocks. Each has one convolution layer, one pool layer, and one batch normalization layer. A single-fully-connected layers layer serves as the classifier. All methods are implemented with PyTorch (Paszke et al., 2019). The maximum training epoch is set to 150. The Adam optimizer with weight decay $5 \times 10^{-4}$ is used. The learning rate for the rest methods is $10^{-2}$. We tune hyperparameters for each method. Moreover, we add another two latest methods, GILE (Qian et al., 2021) and AdaRNN (Du et al., 2021), for comparison in Cross-Person generalization setting.

---

[4]The values in parentheses are the results reported in other papers. They do not mean anything due to different splits, optimizers, settings, etc.

[5]The baselines for UCI-HAR are good enough, so we do not run experiments on it.

**Results**   The results are shown in Table 4, 5, and 6 for three settings, respectively. Our method achieves the best average performance compared to the other state-of-the-art methods in all three settings. In the Cross-Person setting, DIFEX has about $2.8\%, 5.5\%, 1.5\%$ improvement compared to the second-best method for DSADS, USC-HAD, and PAMAP respectively. DIFEX has an improvement with about $2.3\%$ in the Cross-Position setting while it has an improvement with about $7.6\%$ in the Cross-Dataset setting. The results demonstrate DIFEX has a good generalization capability for time-series classification.

Table 4: Classification accuracy on HAR in Cross-Peroson setting. The **bold** items are the best results.

| Dataset | DSADS | | | | | USC | | | | | PAMAP | | | | |
|---|---|---|---|---|---|---|---|---|---|---|---|---|---|---|---|
| Target | 0 | 1 | 2 | 3 | AVG | 0 | 1 | 2 | 3 | AVG | 0 | 1 | 2 | 3 | AVG |
| ERM | 83.11 | 79.30 | 87.85 | 70.96 | 80.30 | 80.98 | 57.75 | 74.03 | 65.86 | 69.66 | 89.98 | 78.08 | 55.77 | 84.44 | 77.07 |
| DANN | 89.12 | 84.17 | 85.92 | 83.38 | 85.65 | 81.22 | 57.88 | 76.69 | 70.72 | 71.63 | 82.18 | 78.08 | 55.39 | 87.26 | 75.73 |
| CORAL | 90.96 | 85.83 | 86.62 | 78.16 | 85.39 | 78.82 | 58.93 | 75.02 | 53.72 | 66.62 | 86.16 | 77.85 | 49.00 | 87.81 | 75.20 |
| Mixup | 89.56 | 82.19 | 89.17 | **86.89** | 86.95 | 79.98 | 64.14 | 74.32 | 61.28 | 69.93 | 89.44 | 80.30 | 58.45 | 87.68 | 78.97 |
| GroupDRO | 91.75 | 85.92 | 87.59 | 78.25 | 85.88 | 80.12 | 55.51 | 74.69 | 59.97 | 67.57 | 85.22 | 77.69 | 56.19 | 84.95 | 76.01 |
| RSC | 84.91 | 82.28 | 86.75 | 77.72 | 82.92 | 81.88 | 57.94 | 73.39 | 65.13 | 69.59 | 87.11 | 76.92 | **60.26** | 87.84 | 78.03 |
| ANDMask | 85.04 | 75.79 | 87.02 | 77.59 | 81.36 | 79.88 | 55.32 | 74.47 | 65.04 | 68.68 | 86.74 | 76.44 | 43.61 | 85.56 | 73.09 |
| SWAD | 89.69 | 76.80 | 86.93 | 80.13 | 83.39 | 82.40 | 56.56 | 71.07 | 69.08 | 69.78 | 85.18 | 74.93 | 54.06 | 81.78 | 73.99 |
| GILE | 81.00 | 75.00 | 77.00 | 66.00 | 74.75 | 78.00 | 62.00 | **77.00** | 63.00 | 70.00 | 83.00 | 68.00 | 42.00 | 76.00 | 67.50 |
| AdaRNN | 80.92 | 75.48 | 90.18 | 75.48 | 80.52 | 78.62 | 55.28 | 66.89 | 73.68 | 68.62 | 81.64 | 71.75 | 45.42 | 82.71 | 70.38 |
| DIFEX | **94.30** | **87.24** | **92.15** | 85.35 | **89.76** | **83.52** | **69.16** | 76.45 | **79.42** | **77.14** | **90.65** | **82.35** | 59.90 | **89.25** | **80.54** |

Table 5: Classification accuracy on DSADS in Cross-Position setting. The **bold** items are the best results.

| Target | 0 | 1 | 2 | 3 | 4 | AVG |
|---|---|---|---|---|---|---|
| ERM | 41.52 | 26.73 | 35.81 | 21.45 | 27.28 | 30.56 |
| DANN | 45.45 | 25.36 | 38.06 | 28.89 | 25.05 | 32.56 |
| CORAL | 33.22 | 25.18 | 25.81 | 22.32 | 20.64 | 25.43 |
| Mixup | 48.77 | **34.19** | 37.49 | 29.50 | 29.95 | 35.98 |
| GroupDRO | 27.12 | 26.66 | 24.34 | 18.39 | 24.82 | 24.27 |
| RSC | 46.56 | 27.37 | 35.93 | 27.04 | 29.82 | 33.34 |
| ANDMask | 47.51 | 31.06 | 39.17 | 30.22 | 29.90 | 35.57 |
| SWAD | 43.83 | 30.18 | 39.35 | 25.38 | 27.58 | 33.26 |
| DIFEX | **49.55** | 32.73 | **41.75** | **33.43** | **34.20** | **38.33** |

Table 6: Accuracy on HAR. The **bold** and underline items are the best and the second-best results.

| Source | Target | ERM | DANN | CORAL | Mixup | GroupDRO | RSC | ANDMask | SWAD | DIFEX |
|---|---|---|---|---|---|---|---|---|---|---|
| USC-HAD,UCI-HAR,PAMAP | DSADS | 26.35 | 29.73 | 39.46 | 37.35 | **51.41** | 33.10 | 41.66 | 28.30 | 46.90 |
| DSADS,UCI-HAR,PAMAP | USC-HAD | 29.58 | 45.33 | 41.82 | 47.39 | 36.74 | 39.70 | 33.83 | 40.99 | **49.28** |
| DSADS,USC-HAD,PAMAP | UCI-HAR | 44.44 | 46.06 | 39.10 | 40.24 | 33.20 | 45.28 | 43.22 | 43.94 | **46.44** |
| DSADS,USC-HAD,UCI-HAR | PAMAP | 32.93 | 43.84 | 36.61 | 23.12 | 33.80 | 45.94 | 40.17 | 16.11 | **52.72** |
| AVG | - | 33.32 | 41.24 | 39.25 | 37.03 | 38.79 | 41.01 | 39.72 | 32.33 | **48.83** |

Moreover, we have some more insightful findings.

1. *Can the simple alignments always bring benefits?* Obviously, the answer is no. Although CORAL can bring improvements compared to ERM for DSADS in the Cross-Person setting and in the Cross-Dataset setting, it performs worse than ERM for the other benchmarks.

2. *Different alignments bring different effects.* It seems that DANN performs better than CORAL, which inspires us that we can be able to replace CORAL with DANN to capture mutually-invariant features in the future work.

3. *Some methods proposed for visual classification initially also work for time series.* From Table 4, 5, and 6, we can see that RSC brings improvements in most circumstances although it was initially designed for visual classification, which demonstrates there may exist commonness between these

two different modalities. However, compared to remarkable improvements for visual classification, the improvements for time-series data are not significant anymore. It illustrates that these methods may be not general enough.

4. *The same method for different DG benchmarks may have completely different performances.* ERM, the baseline without any generalized techniques, sometimes performs better than other state-of-the-art methods, e.g., for USC in the Cross-Person setting while it performs worse than others sometimes. This phenomenon illustrates that domain generalization is a hard task and we need to design different methods for various situations. In this condition, our method that can achieve the best performance almost on all tasks is inspiring.

5. *For data with fewer raw channels and for more difficult benchmarks, learning more diverse features brings better improvements.* In the Cross-Person setting, USC-HAD is the most difficult benchmark containing only 6 channels. In this situation, how to exploit more diversity and useful features is more important. Thus, our method with both internally-invariant and mutually-invariant features can bring much larger improvements. A similar phenomenon exists in the Cross-Dataset setting.

### 4.3 EMG gesture recognition

To further prove the superiority of our methods on time-series data, we also validate our method on a more challenging benchmark, EMG for gestures Data Set Lobov et al. (2018) which contains raw EMG data recorded by MYO Thalmic bracelet[6]. Electromyography (EMG) is based on bioelectric signals and is a common type of time-series data used in many fields, such as healthcare and entertainment. For the dataset, the bracelet is equipped with eight sensors equally spaced around the forearm and eight sensors simultaneously acquire myographic signals when 36 subjects performed

Table 7: Results on EMG in Cross-Person setting.

| Target | 0 | 1 | 2 | 3 | AVG |
|---|---|---|---|---|---|
| ERM | 64.08 | 58.05 | 52.51 | 57.29 | 57.98 |
| DANN | 60.68 | 70.40 | 64.06 | 52.39 | 61.88 |
| CORAL | 62.97 | 73.70 | 71.53 | 70.17 | 69.59 |
| Mixup | 59.21 | 66.54 | 61.54 | 64.62 | 62.98 |
| GroupDRO | 63.62 | 67.64 | 72.19 | 67.99 | 67.86 |
| RSC | 65.02 | 75.19 | 69.86 | 61.25 | 67.83 |
| ANDMask | 59.33 | 66.15 | 71.83 | 65.09 | 65.60 |
| SWAD | 42.02 | 57.06 | 53.89 | 48.73 | 50.42 |
| DIFEX | **71.83** | **82.8** | **76.91** | **75.84** | **76.84** |

series of static hand gestures. The dataset contains  40,000 − 50,000 recordings in each column with 7 classes and we select 6 common classes for our experiments. We randomly divide 36 subjects into four domains without overlapping and each domain contains data of 9 persons. The baseline methods and implementation details are similar to those for HAR. Experiments are done in three random trials, thus mitigating influences of unfair splits. EMG data comes from bioelectric signals, and thereby it is affected by many factors, such as environments and devices, which means the same person may generate different sensor data even when performing the same activity with the same device at a different time or with the different devices at the same time. Therefore, this benchmark is more challenging.

Table 7 shows that our method achieves the best average performance and is **7.25**% better than the second-best method. Just similar to results mentioned above, alignments among domains can bring improvements in most circumstances while only traditional alignments are not enough, which proves the need of both internally-invariant features and mutually-invariant features. Moreover, from the first task in Table 7, we see that sometimes simple domain alignments may deteriorate performance. But ours performs the best on all tasks.

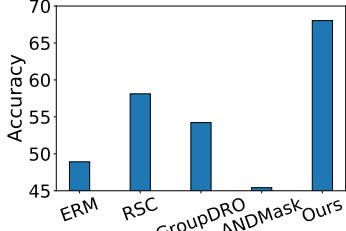

Figure 4: Results on USC-HAD for single-person DG.

### 4.4 Single domain generalization

In this section, we perform single domain generalization to demonstrate superiority of our method.

**USC-HAD**   We randomly choose two subjects from USC-HAD and each subject serves as one domain. As shown in Figure 4, DIFEX achieves the best performance with over 9% improvements compared to four other state-of-the-art methods. It demonstrates DIFEX can make full use of both internally and mutually domain-invariant features. To sum up, DIFEX is effective in both image and time-series benchmarks with multiple and single domains, indicating that it is a general approach for domain generalization.

---

[6]For more details about EMG, please refer to `https://archive.ics.uci.edu/ml/datasets/EMG+data+for+gestures`.

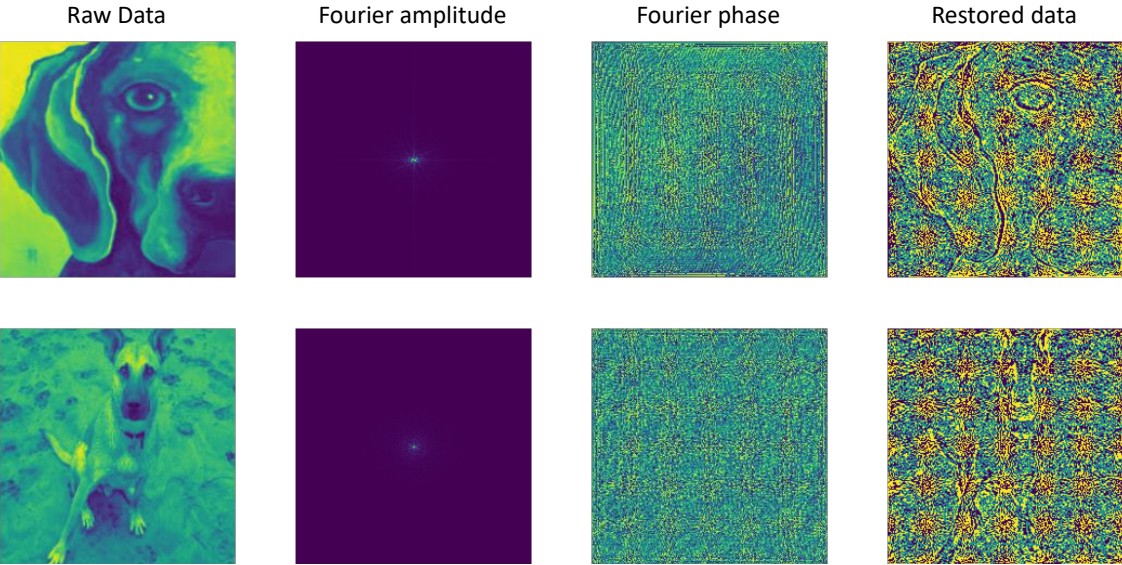

Figure 5: Fourier features as internally-invariant representations for images.

Table 8: Accuracy on PACS. The **bold** and underline items are the best and the second-best results.

| Source | Target | ERM | Mixup | GroupDRO | RSC | ANDMask | DIFEX |
|--------|--------|-------|-------|----------|-------|---------|--------|
| S | A | 35.50 | 36.67 | 38.53 | 36.57 | 35.50 | **46.68** |
| A | C | 57.25 | 58.49 | 59.94 | 62.50 | 57.25 | **64.46** |
| C | P | 81.86 | 82.87 | 84.55 | 84.49 | 81.86 | **86.17** |
| P | S | 31.48 | 32.25 | 34.11 | 33.16 | 31.48 | **56.81** |
| AVG | - | 51.52 | 52.57 | 54.28 | 54.18 | 51.52 | **63.53** |

**PACS**   We choose two datasets from PACS each time. We treat one selected dataset as the source domain and the other as the target. As shown in Table 8, our method achieves the best performance with over 9% improvements on average compared to five latest state-of-the-art methods. In this situation, GroupDRO shows its ability of generalization since it is designed for out-of-distribution without many source domains while RSC also shows an acceptable performance since it is a general method. The results demonstrate our method can perform well on single domain generalization for image classification.

## 4.5   Analysis

**Ablation Study**   We perform ablation study in this section.

*Does it work if using internally-invariant or mutually-invariant features alone?*   As it is shown in Figure 6, directly utilizing mutually-invariant features brings a slight improvement compared to ERM while only utilizing internally-invariant features distilled from a teacher net has the same performance as ERM. Combining both two kind of features bring a much better performance. This demonstrates that only internally-invariant or mutually-invariant features may be not discriminative enough.

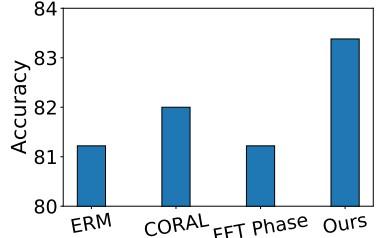

Figure 6: Ablation study on the PACS dataset.

*Do three parts of Eq. 7 all work?*   As shown in Table 9, without one part still has an improvement compared to ERM. Compared to mutually-invariant features and exploration, internally-invariant features play a slightly less important role. And combining three parts can bring the best performance, showing that each part is essential for generalization.

Table 9: Ablation study.

| Benchmark | PACS with multiple sources | | | | | HAR in Cross-Dataset | | | | | PACS with single source | | | | |
|---|---|---|---|---|---|---|---|---|---|---|---|---|---|---|---|
| Target | A | C | P | S | AVG | D | U | H | P | AVG | A | C | P | S | AVG |
| ERM | 77.00 | 74.53 | 95.51 | 77.86 | 81.22 | 26.35 | 29.58 | 44.44 | 32.93 | 33.32 | 35.50 | 57.25 | 81.86 | 31.48 | 51.52 |
| w./o Intern. | 80.66 | 75.51 | 95.33 | 78.19 | 82.42 | **46.90** | 47.26 | 45.76 | 51.35 | 47.82 | 41.75 | 63.52 | 84.91 | 50.42 | 60.15 |
| w./o Mutual. | 80.47 | 75.60 | 95.33 | 78.06 | 82.36 | 42.61 | 44.12 | 45.12 | 46.49 | 44.58 | 43.16 | 63.01 | 84.91 | 41.77 | 58.21 |
| w./o Exp. | **80.86** | 75.85 | 95.33 | 78.14 | 82.55 | 38.21 | 48.63 | 45.88 | 51.25 | 45.99 | 43.02 | 63.99 | 85.75 | 44.90 | 59.42 |
| DIFEX | **80.86** | **77.60** | **95.57** | **79.49** | **83.38** | **46.90** | **49.28** | **46.44** | **52.72** | **48.83** | **46.68** | **64.46** | **86.17** | **56.81** | **63.53** |

**Motivation Examples for Visual Images**   We have shown an example that some Fourier features can be viewed as internally-invariant representations for time-series data, and we give an example for visual data. As shown in Figure 5, Fourier features for images are similar to features for sensors. Two figures in the second column demonstrate that Fourier amplitude may be useless for classification. Two figures in the last column are data restored with only Fourier phase. They illustrate that Fourier phase contains most of classification information since we can vaguely see dogs while it still loses some information compared to the original data since the dogs are not clear compared to original raw data (two figures in the first column). [7]

**Parameter Sensitivity**   There are mainly three hyperparameters in our method: $\lambda_1$ for distillation knowledge from a teacher net with Fourier phase information, $\lambda_2$ for the CORAL alignment loss, and $\lambda_3$ for the exploration loss. We evaluate the parameter sensitivity of our method in Figure 7 where we change one parameter and fix the other to record the results. From these results, we can see that our method achieves better performance near the optimal point, demonstrating that our method is insensitive to some hyperparameter choices to some extent. We also note that $\lambda_3$ for exploration loss is a bit sensitive which may need attention in real applications.[8]

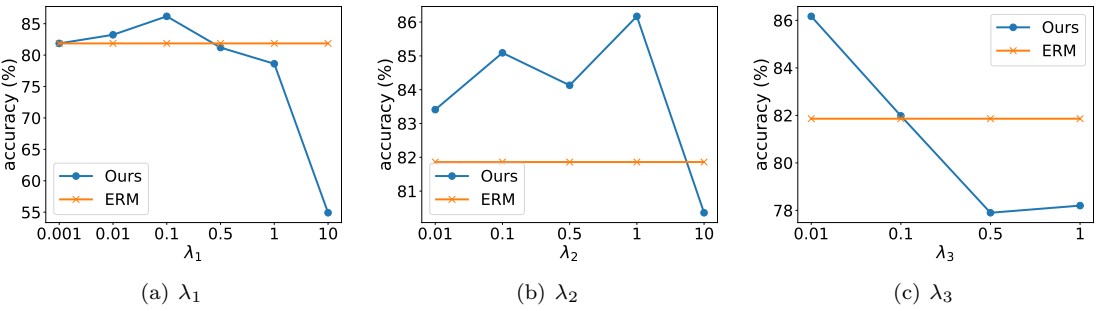

(a) $\lambda_1$          (b) $\lambda_2$          (c) $\lambda_3$

Figure 7: Parameter sensitivity (C→P in PACS for single domain generalization).

## 5   Conclusion

In this paper, we propose DIFEX to learn both internally-invariant and mutually-invariant features for domain generalization. DIFEX utilizes an inheritance and exploration framework to combine internally-invariant features distilled from a teacher net with Fourier phase information and mutually-invariant features obtained from domain alignments. Extensive experiments on both image and time-series classification demonstrate the superiority of DIFEX.

In the future, we plan to combine more internally-invariant features and mutually-invariant features for better generalization. Moreover, we may utilize normalization techniques or some other distance to guide explorations for more stable optimization.

---

[7]Although restored data in Fig.6 have many checkboard artifacts, they have little influence on performance. We first utilize a teacher network to learn useful information for classification. And then we extract internally-invariant representations with the help of the teacher net which has filtered noisy information. In this step, we also utilize the classification loss to reduce the influence of useless information.

[8]When $\lambda_3$ is smaller, e.g. $\lambda = 0$, the performance will drop, as shown in Table 9.

## 6 Acknowledgments

This work is supported by the National Key Research and Development Plan of China (No. 2019YFB1404703), Natural Science Foundation of China (No. 61972383), Science and Technology Service Network Initiative, Chinese Academy of Sciences (No. KFJ-STS-QYZD-2021-11-001), and the Research Grant Council (RGC) of Hong Kong through Early Career Scheme (ECS) under the Grant 21200522 and CityU Applied Research Grant (ARG) 9667244.

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

## A  More details about DIFEX

### A.1  Novelty

Learning the private and shared representations, or disentanglement, is a quite popular research direction for domain adaptation/generalization. However, the novelty is never the disentanglement idea itself, but how we can achieve such representation separation. From this point of view, our method is novel since it is the first work that splits the representations into internally and mutually invariant aspects. Most of the existing disentanglement works are based on a variational autoencoder structure, which is totally different from ours.

Fourier features are also popular in recent years, but the problem is that the novelty lies in how to adopt Fourier features for DG. To this end, our approach is totally different from existing research:

- Please note that Xingchen et al. (2021) is not for a traditional DG setting. The model changes when testing. Xu et al. (2021) utilized Fourier amplitude for data augmentation. Our method directly utilizes knowledge distillation to extract internally-invariant features, i.e. Fourier phase information. It is no doubt that ours has a totally different implementation.

- Applying the high-level Fourier features is only part of our method. It is for internally-invariant features. Our method also includes mutually-invariant feature learning. These concepts and the combination are novel. Moreover, we also include exploration. To the best of our knowledge, few methods pay attention to the exploration among different types of features for diversity.

Overall, our method is novel enough.

### A.2 Compare to other Fourier-based methods, e.g. FACT

There are several existing approaches exploiting Fourier-based features for DG, as discussed in related work section. More specifically, although FACT Xu et al. (2021) and ours share similar ideas, we are totally different in methodology:

- FACT utilizes amplitude Mixup to perform data augmentation and its teacher-student framework is for consistency alignment. It can be seen as an adaptation of the data augmentation technique and the model optimization technique.

- Although our DIFEX also tries to utilize Fourier phase information, we focus on the feature extraction. And our teacher network directly extracts the Fourier phase information for the student network to learn.

- Some other methods utilizing Fourier for domain generalization are mentioned in Section 2.3 of the main paper. However, none of them directly utilizes the Fourier phase to guide the part of feature learning (internally invariant features).

We offer some comparison results between our method and FACT in Figure 8. Obviously, our method achieves the better results, especially in time-seriese benchmarks.

### A.3 How the mutually-invariant and the internally-invariant features are extracted.

As shown in Figure 2, this operation is done by directly splitting the last layer of the feature network into two parts: one for mutually-invariant and the other for internally-invariant features, respectively. This can be easily implemented by the neural network design. Then, we utilize the learning goals (i.e. distillation, alignment, and exploration) to guide the feature extraction process. The feature extractor network, except for the last layer, is shared by these two parts (the blue block in the figure). The summation operation (i.e., the circled '+' sign) in the figure denotes concatenating two parts. We believe that utilizing independent networks for these two parts respectively can bring better performance. However, it may be unfair since it utilizes a larger feature network.

## B Implementation details

We try our best to ensure fair comparisons. As mentioned in the main paper, we utilize the same backbone, the same optimizer, the same splits, and some other same tricks, e.g. data transformation (The epoch is so large that each method can reach convergence.). We tune hyperparameters for each method and select the best to compare (we even tune some base hyperparameters, e.g. lr, for some methods but we report the mainstream of the base hyperparameters.).

Moreover, note that DomainBed is a popular benchmark but not the only one. Domainbed utilizes Adam and totally random parameters, which are not suitable for some methods. Therefore, lots of methods, e.g. FACT, do not utilize it. In addition, utilizing the results reported in the original paper are also not suitable. Different tricks, different data splits, and even different initial states can have large influences on the final results (As shown in DomainBed, in the totally same and random environment, some methods decline seriously.). We try to get a balance between totally random and good performance. We believe ours can achieve better performances with more careful tuning. But we focus on methods and fairness.

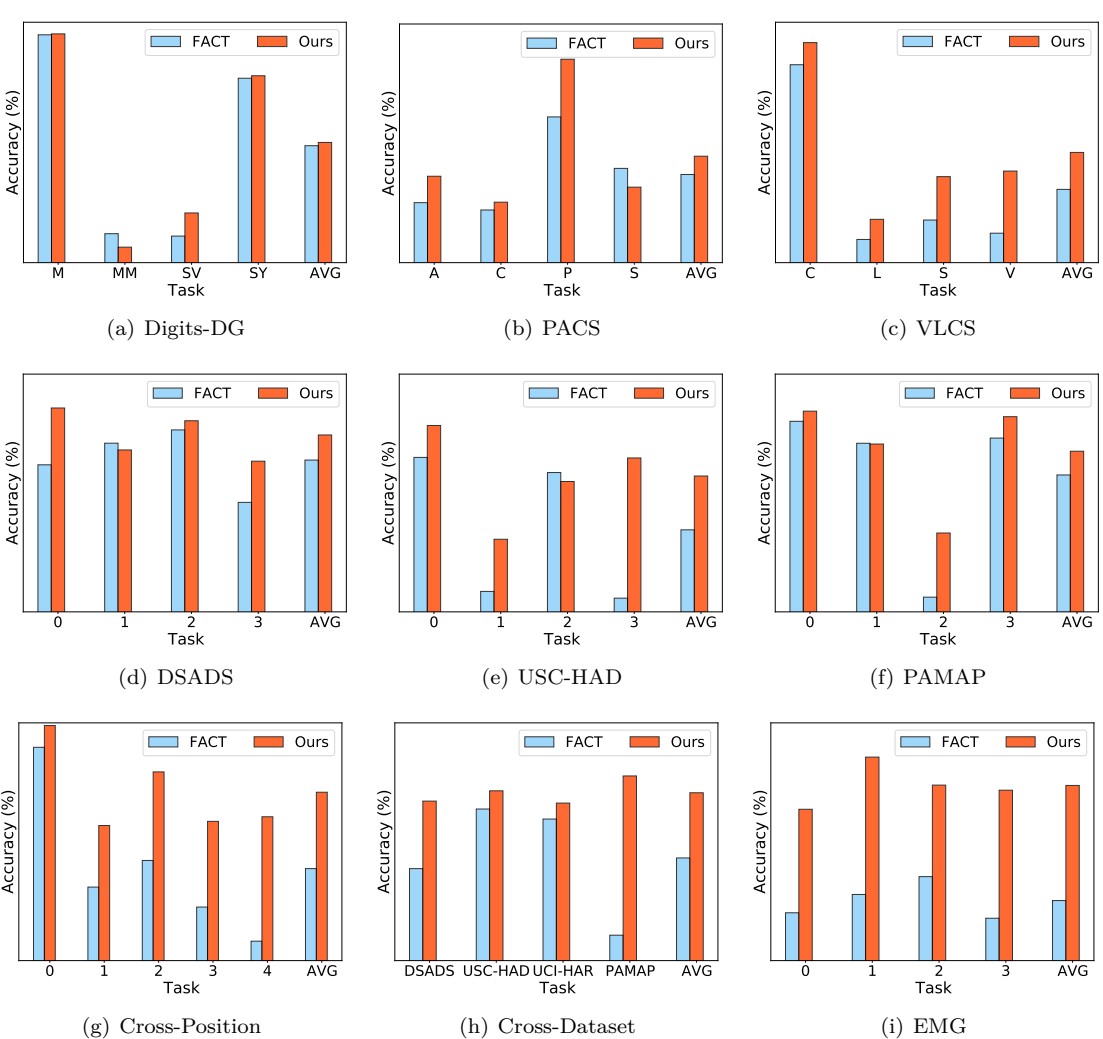

Figure 8: Comparison results between our method and FACT.

## C   Limitations

Table 10: Classification accuracy with different exploration distance computation ways.

| Dataset | Target | A | C | P | S | AVG | |
|---|---|---|---|---|---|---|---|
| PACS | $L2$ | 80.86 | 77.60 | 95.57 | 79.49 | 83.38 | |
| | norm-$L1$ | 81.30 | 78.41 | 96.17 | 80.02 | 83.98 | |
| Dataset | Target | 0 | 1 | 2 | 3 | AVG | |
| PAMAP | $L2$ | 90.65 | 82.35 | 59.90 | 89.25 | 80.54 | |
| | norm-$L1$ | 90.85 | 82.51 | 63.71 | 89.32 | 81.60 | |
| Dataset | Target | DSADS | USC-HAD | UCI-HAR | PAMAP | AVG | |
| Cross-Dataset | $L2$ | 46.90 | 49.28 | 46.44 | 52.72 | 48.83 | |
| | norm-$L1$ | 56.25 | 49.36 | 46.35 | 56.43 | 52.10 | |
| Dataset | Target | 0 | 1 | 2 | 3 | 4 | AVG |
| Cross-Position | $L2$ | 49.55 | 32.73 | 41.75 | 33.43 | 34.20 | 38.33 |
| | norm-$L1$ | 50.87 | 33.85 | 43.49 | 34.71 | 33.46 | 39.28 |

We have mentioned in the main paper that exploring with $L2$ distance may bring a difficulty to optimize ($L2$ can be infinity). Therefore, we try another computation way, a normalization distance (norm-$L1$),

$$L_{exp1}(\mathbf{z_1}, \mathbf{z_2}) = -||\frac{\mathbf{z_1}}{||\mathbf{z_1}||_2} - \frac{\mathbf{z_2}}{||\mathbf{z_2}||_2}||_1. \tag{9}$$

The results are shown in Table 10 and we can see that norm-$L1$ can bring improvements, especially for difficult problems, e.g. Cross-Dataset[9]. In the future, we may try some other distance, such as cosine.

## D   Visualization study

We present some visualizations to show the reasons behind the rationales of our method. As shown in Figure 9, we perform t-SNE to show the learned features of training data and target data directly. Figure 9(a) demonstrates that ERM without any generalization operations leads that the learned features of test data have a different distribution from the learned features of training data and target data are hard to classify. Figure 9(b) and Figure 9(c) illustrate that learning with alignments can alleviate the above problem but some classes still suffer from the same issue, especially for CORAL. Since mutually-invariant features are learned with the same technique as CORAL, a similar phenomenon exists in Figure 9(f), which again emphasizes the necessity of the other techniques. And Figure 9(e) proves that internally-invariant features may make sense for generalization. Figure 9(d) demonstrates that our methods can further alleviate distribution shifts compared to traditional methods with domain alignments, and thereby it can bring better performance. Moreover, with the last fully-connect layer, we can assign different weights to internally-invariant features and mutually-invariant features for better performance in different situations.

---

[9]Please note that our method with $L2$ has achieved best results compared to other methods and normalization techniques are mainly for better performance and easier optimization

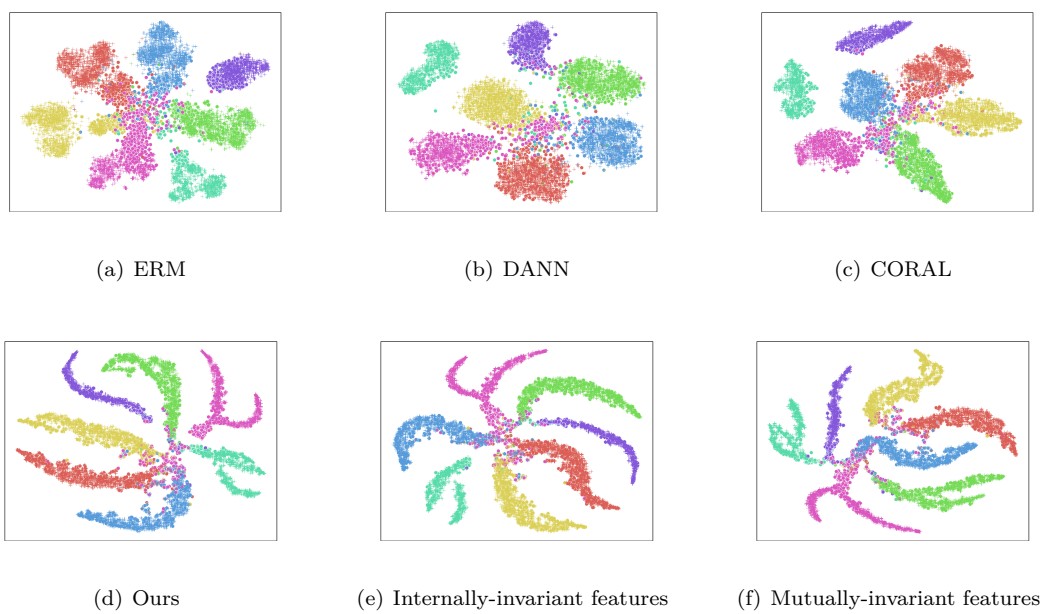

Figure 9: Visualization of the t-SNE embeddings of learned feature spaces for PACS with different methods. Different colors correspond to different classes and different shapes correspond to different domains. Circle and plus correspond to test data and training data respectively. *Best viewed in color and zoom in.*

