# OpenReview forum: "Domain-invariant Feature Exploration for Domain Generalization"
_TMLR — Accepted by TMLR_

### Review · Reviewer_gsxo · 2022-06-04

**Summary Of Contributions:**

This paper aims to solve the domain generalization problem by learning domain-invariant features from two aspects, including using the knowledge distillation with Fourier transform to capture internal invariant features and using correlation alignment to capture mutual invariant features. The proposed method is validated on a time-series dataset and three DG benchmark datasets.

**Broader Impact Concerns:**

This paper has no broader impact concerns.

**Requested Changes:**

Some more detailed comments are as follows:

- Figure1 gives a toy example on 1-d data, how’s the quality of using phase to reconstruct data on 2-d images? Is there any information loss by solely using the phase data? Besides, the restored data in Fig.6 have many checkboard artifacts, which may introduce noisy patterns.

- The motivation for aligning correlations to learn mutually invariant features are not clear. Why can aligning correlations make the feature mutually invariant? Here needs to be further clarified.

- In Eq. (5), it is not clear how to calculate the covariance matrix C^i.

- In the parameter sensitivity study, the accuracy is high with a small value for \lambda_3. What is the performance if the \lambda_3 is smaller than 0.01?

**Strengths And Weaknesses:**

Pros:

- This paper tackles an important problem of domain generalization, in which target data is not available for training.
- The manuscript is well organized.
- The idea of learning invariant features for both internal and mutual-side is interesting.
- The method is evaluated on the time-series data.

Cons:

- Some methodology details need to be further clarified (see below requested changes)
- The comparison lacks some strong DG methods, e.g., Mixstyle[1], SWAD[2], etc. Besides, this paper[3] applies Fourier transformation for domain generalization tasks, comparison and analysis with this method would make the experiments more convincing.
- The experiment results are much lower than the results reported in the DomainBed. And the performance gaps for different methods are inconsistent. Is there any explanation?
- It is not clear how the proposed method can perform single domain generalization. How is the proposed cross-domain correlation alignment applied? Besides, there are methods specifically designed for the single domain generalization task; the current comparison using methods not specifically designed for the single domain may not be fair.

[1] Zhou, Kaiyang, Yongxin Yang, Yu Qiao, and Tao Xiang. "Domain generalization with mixstyle." ICLR (2021).

[2] Cha, Junbum, Sanghyuk Chun, Kyungjae Lee, Han-Cheol Cho, Seunghyun Park, Yunsung Lee, and Sungrae Park. "Swad: Domain generalization by seeking flat minima." Advances in Neural Information Processing Systems 34 (2021).

[3] Xu, Qinwei, Ruipeng Zhang, Ya Zhang, Yanfeng Wang, and Qi Tian. "A fourier-based framework for domain generalization." In Proceedings of the IEEE/CVF Conference on Computer Vision and Pattern Recognition, pp. 14383-14392. 2021.

---

> ### Author Response · Authors · 2022-06-08
> **Response for Reviewer gsxo (1)**
>
> Thank you for your professional and constructive feedback. We are very glad that you acknowledge our technical novelty, paper writing, and comprehensive evaluations. Now we answer your concerns below. NOTE: we will revise the paper accordingly when all reviews are finished. Please stay tuned.
> 1. More strong DG comparison methods.
>
> We only choose some popular and general methods for comparison. Mixstyle [1] and FACT [3] are designed for the CV area while SWAD [2] is a training enhancement technology that can be embedded in existing methods. To further alleviate your concerns, we re-implement these three methods and compared them with our methods.
>
> The results are shown in the following table. We can see that our method achieves the best performance on all nine benchmarks. Since these strong comparison methods are mainly designed for the visual field, they perform seriously worse than ours in time-series benchmarks even if we carefully tune their hyperparameters.
>
> | Digits-DG      | M     | MM      | SV      | SY    | AVG       | DSADS     | 0     | 1     | 2     | 3     | AVG       |
> |----------------|-------|---------|---------|-------|-----------|-----------|-------|-------|-------|-------|-----------|
> | Mixstyle       | 96.58 | 60.32   | 55.87   | 87.92 | 75.17     | Mixstyle  | 90.18 | 78.29 | 87.76 | 75.13 | 82.84     |
> | FACT           | 97.65 | 60.42   | 59.97   | 89.52 | 76.89     | FACT      | 84.74 | 88.38 | 90.61 | 78.42 | 85.54     |
> | SWAD+ERM       | 97.37 | 55.77   | 59      | 87.92 | 75.02     | SWAD+ERM  | 89.69 | 76.8  | 86.93 | 80.13 | 83.39     |
> | Ours           | 97.82 | 57.9    | 64.3    | 89.98 | **77.5**  | Ours      | 94.3  | 87.24 | 92.15 | 85.35 | **89.76** |
> | PACS           | A     | C       | P       | S     | AVG       | USC       | 0     | 1     | 2     | 3     | AVG       |
> | Mixstyle       | 83.11 | 78.37   | 93.65   | 76.23 | 82.84     | Mixstyle  | 75.62 | 53.6  | 78.21 | 57.55 | 66.25     |
> | FACT           | 77.54 | 76.62   | 88.32   | 81.85 | 81.08     | FACT      | 79.49 | 62.58 | 77.58 | 61.73 | 70.35     |
> | SWAD+ERM       | 81.93 | 76.45   | 94.67   | 76.79 | 82.46     | SWAD+ERM  | 82.4  | 56.56 | 71.07 | 69.08 | 69.78     |
> | Ours           | 80.86 | 77.6    | 95.57   | 79.49 | **83.38** | Ours      | 83.52 | 69.16 | 76.45 | 79.42 | **77.14** |
> | VLCS           | C     | L       | S       | V     | AVG       | PAMAP     | 0     | 1     | 2     | 3     | AVG       |
> | Mixstyle       | 95.83 | 65.02   | 68.25   | 72.33 | 75.36     | Mixstyle  | 82.89 | 76.15 | 53.48 | 79.21 | 72.93     |
> | FACT           | 92.93 | 63.86   | 67.09   | 64.9  | 72.19     | FACT      | 88.09 | 82.55 | 43.71 | 83.86 | 74.55     |
> | SWAD+ERM       | 94.56 | 61.78   | 67.82   | 67.89 | 73.01     | SWAD+ERM  | 85.18 | 74.93 | 54.06 | 81.78 | 73.99     |
> | Ours           | 96.61 | 67.21   | 74.31   | 75.24 | **78.34** | Ours      | 90.65 | 82.35 | 59.9  | 89.25 | **80.54** |
> | Cross-Dataset  | DSADS | USC-HAD | UCI-HAR | PAMAP | AVG       | EMG       | 0     | 1     | 2     | 3     | AVG       |
> | Mixstyle       | 7.18  | 39.37   | 37.96   | 36.04 | 30.14     | Mixstyle  | 64.26 | 71.72 | 68.48 | 70.53 | 68.75     |
> | FACT           | 31.28 | 45.07   | 42.75   | 15.88 | 33.75     | FACT      | 50.06 | 53.91 | 57.66 | 48.91 | 52.63     |
> | SWAD+ERM       | 28.3  | 40.99   | 43.94   | 16.11 | 32.33     | SWAD+ERM  | 42.02 | 57.06 | 53.89 | 48.73 | 50.42     |
> | Ours           | 46.9  | 49.28   | 46.44   | 52.72 | **48.83** | Ours      | 71.83 | 82.8  | 76.91 | 75.84 | **76.84** |
> | Cross-Position | 0     | 1       | 2       | 3     | 4         | AVG       |       |       |       |       |           |
> | Mixstyle       | 36.32 | 25.13   | 35.27   | 25.93 | 19.73     | 28.48     |       |       |       |       |           |
> | FACT           | 45.89 | 22.37   | 26.85   | 19.01 | 13.27     | 25.48     |       |       |       |       |           |
> | SWAD+ERM       | 43.83 | 30.18   | 39.35   | 25.38 | 27.58     | 33.26     |       |       |       |       |           |
> | Ours           | 49.55 | 32.73   | 41.75   | 33.43 | 34.2      | **38.33** |       |       |       |       |           |
>
> 2. Compared to DomainBed.
>
> We do not implement our method with DomainBed. DomainBed may require lots of computation resources. Their public results are based on ResNet50 while ours are based on ResNet18. And we have some implementation differences from DomainBed, e.g. different optimizers.

---

> > ### Comment · Reviewer_gsxo · 2022-07-12
> > **Response to feedback**
> >
> > Thanks for the authors' effort on major revision. All my previous questions have been addressed. Please be noted to incorporate the extended contents in final version.

---

> ### Author Response · Authors · 2022-06-08
> **Response for Reviewer gsxo (2)**
>
> 3. How to perform single domain generalization.
>
> We have mentioned it in Section 3.4. We can give random domain labels to perform domain alignment when training one domain. Specifically, we label the first half of a batch as domain 0 and label others as domain 1.
>
> In addition to some unified DG methods, there are only a few methods specially designed for single domain generalization. To alleviate your concerns, we compare our method with UMGUD [4]. In the following table, we can see that our method still achieves better performance.
>
> | USC-HAD | AVG       |       |       |       |           |
> |---------|-----------|-------|-------|-------|-----------|
> | UMGUD   | 53.27     |       |       |       |           |
> | Ours    | **68.03** |       |       |       |           |
> | PACS    | A         | C     | P     | S     | AVG       |
> | UMGUD   | 37.21     | 60.12 | 82.95 | 37.79 | 54.52     |
> | Ours    | 46.68     | 64.46 | 86.17 | 56.81 | **63.53** |
>
> [4] Qiao, Fengchun, and Xi Peng. "Uncertainty-guided model generalization to unseen domains." Proceedings of the IEEE/CVF Conference on Computer Vision and Pattern Recognition. 2021.
>
> #### Requested changes:
>
> 4. Quality of reconstructed data.
>
> In the main paper, Fig.6 in Section 4.5 gives an example on 2-d images. Yes, there is some information loss by solely using the phase data. And it is the reason why we also need mutually-invariant features. Although restored data in Fig.6 have many checkboard artifacts, they have little influence on performance. We first utilize a teacher network to learn useful information for classification. And then we extract internally-invariant representations with the help of the teacher net which has filtered noisy information. In this step, we also utilize the classification loss to reduce the influence of useless information.
>
> 5. Why aligning correlations can make the feature mutually invariant.
>
> Mutually-invariant features focus on harnessing the cross-domain transferable knowledge
> that is mined from different distributions. We think alignments among sources can bring common knowledge of sources. And the correlation alignment approach, i.e. CORAL[5], is a popular technique for DA and DG.
> [5] Sun, Baochen, and Kate Saenko. "Deep coral: Correlation alignment for deep domain adaptation." European conference on computer vision. Springer, Cham, 2016.
>
> 6. How to calculate the covariance matrix.
>
> We give the calculation formula following [5].
> $C = \frac{1}{n-1} (\mathbf{X}^T\mathbf{X} - \frac{1}{n}(\mathbf{1}^T\mathbf{X})^T(\mathbf{1}^T\mathbf{X}))$
>
> 7. When \lambda_3 has a small value.
>
> The results are shown in the following table. Note that our method prefers little \lambda_3 for this task. Moreover, we have demonstrated our method with the exploration loss (\lambda_3) performs better in ablation study.
>
> | lambda3 | 0     | 0.001 | 0.01  | 0.1   | 0.5  | 1    |
> |---------|-------|-------|-------|-------|------|------|
> | Ours    | 85.75 | 85.92 | 86.17 | 81.98 | 77.9 | 78.2 |

---

### Review · Reviewer_B7sJ · 2022-06-08

**Summary Of Contributions:**

The article presents a new framework for domain generalization (DG) based on feature alignment, named DIFEX. DIFEX is a teacher-student framework, where the teacher receives as input the phase of Fourier transformed images (preserving semantic rather than style information). The student is then trained to extract the same, style-agnostic features, but from non-transformed images.  To overcome a possible deterioration of the extracted features and exploit cross-domain information, an additional alignment loss is applied to the second-order statistics of features across domains (i.e. the CORAL loss of Sun & Saenko 2016), and the distance between features used in the first and second loss terms is maximized to avoid redundancy of the two terms. Experimental results show that the model achieves good results across different DG benchmarks on images and time-series data.

**Broader Impact Concerns:**

None.

**Requested Changes:**

Following on the previous section, I deem as important points for revision:
1. Improving the clarity of the manuscript for what concerns the definitions (point 6 above) and methodological components (point 1 above).
2. Expand the comparison with related works (point 2 above) and provide the performance of other approaches when hyperparameters are tuned for their specific design choices, at least when available already in the literature (see point 4 above).
3. Provide more details/clarifications (point 5) and expand the ablation analysis (point 3) on Fig.7.

**Strengths And Weaknesses:**

**Strengths**:
- The approach is interesting, coupling standard alignment techniques (i.e. CORAL) with Fourier-based ones (e.g. FACT). Exploring potential redundancy between the two is an interesting topic of research that might shed light on how to perform domain invariant learning on DG.
- The analysis in section 3.2 well motivates the benefits and drawbacks of using only Fourier transformed inputs.

**Weaknesses**:
1. In the method it is unclear how the mutually invariant and the internally invariant features are extracted. The reason is that while Fig.2 shows two different features (i.e. yellow and green blocks), Fig. 3 shows that there is no particular feature extractor for the internally invariant one. Thus,  in Eq. (6) it is unclear how $\mathbf{z}_2$ is obtained and how it differs from $\mathbf{z}_1$, given that they share the same feature extractor. Note also that Fig. 2 hints at the presence of a summation operation between the two, which is not reflected in the training/inference as described in Eq.(7) and Eq.(8). It is of utmost importance that these aspects are revised to avoid ambiguities and make DIFEX clear to the reader.
1. The approach is very similar in spirit to FACT (Xu et al. 2021) where also a teacher-student framework is proposed for DG. Also in FACT, the teacher receives as input Fourier-transformed images, while the student their original version, and a consistency loss is minimized between their extracted features. While Section 2.3 describes FACT,  it is not mentioned the similarity between FACT and DIFEX, and the former is not reported in the experimental analysis. Section 2.3 should better highlight the differences between DIFEX and FACT (as well as existing approaches using Fourier transform), and the experiments should include this baseline to clearly show the advantages and disadvantages of these two related frameworks. Note that FACT reports an average accuracy on PACS of 84.51, which is slightly higher than the one reported for DIFEX (i.e. 83.38, Table 2).
2. According to Fig.7c, the lower the weight is given to the exploration and the higher the performance of the model, reducing its practical impact. This contrasts with the message of the title (stressing this component). It would be helpful to report the performance for negative values of $\lambda_3$, to show whether or not exploring actually may even improve the performance (something suggested by the trend of Fig.7c).
3. According to section 4.1, the experimental comparisons with previous approaches to image classification are mainly held out by re-training each model with the same set of shared hyperparameters. While this is a good effort toward decoupling optimization from model choices, still each of the previous approaches might need specific hyperparameters due to, e.g. different convergence times, etc. This may lead a reader to wonder whether the experimental comparison is actually fairer in this way or whether the performance of the other approaches may be reduced by this. Given that the purpose of DomainBed (used to implement previous models, as of Section 4) is exactly to provide a fair benchmark of comparison, I believe that a fairer evaluation would have been performed by either i) reporting the same results of previous works as reported in their respective papers (or DomainBed, if available), since this would have ensured tuned hyperparameters per approach; or ii) evaluating DIFEX on the DomainBed benchmark (to ensure the same evaluation ground). While the second option might be expensive in terms of resources, the first one (tuning the hyperparameters per DIFEX using the available validation splits) would still be viable.
4. In Fig. 7, the numbers are higher than what is reported in the tables (i.e. over 85% vs 83.38% in Table 9). Why is this the case?
5. From the abstract, it is not trivial to get the meaning of internal and mutual invariance. I would consider revising it to clarify these concepts.

---

> ### Author Response · Authors · 2022-06-11
> **Response for Reviewer B7sJ (1)**
>
> Thank you for your professional and constructive feedback. We are very glad that you acknowledge our approach, analysis, and comprehensive experiments.
>
> Most of your concerns are with the *clarity of the method and comparison with other related works*, which we address as follows.
>
> 1. How the mutually-invariant and the internally-invariant features are extracted.
>
> This operation is done by directly splitting the last layer of the feature network into two parts: one for mutually-invariant and the other for internally-invariant features, respectively. This can be easily implemented by the neural network design. Then, we utilize the learning goals (i.e. distillation, alignment, and exploration) to guide the feature extraction process. The feature extractor network, except for the last layer, is shared by these two parts (the blue block in the figure). The summation operation (i.e., the circled `+' sign) in the figure denotes concatenating two parts. We will further revise our description in the next version of the paper.
>
> Moreover, we believe that utilizing independent networks for these two parts respectively can bring better performance. However, it may be unfair since it utilizes a larger feature network.
>
> 2. Comparison with other Fourier-based approaches such as FACT.
>
> There are several existing approaches exploiting Fourier-based features for DG, as discussed in related work section. More specifically, although FACT and ours share similar ideas, we are **totally different** in methodology:
> - FACT utilizes *amplitude Mixup* to perform *data augmentation* and its teacher-student framework is for *consistency alignment*. It can be seen as an adaptation of the data augmentation technique and the model optimization technique.
> - Although our DIFEX also tries to utilize Fourier phase information, we focus on the *feature extraction*. And our teacher network directly extracts the Fourier phase information for the student network to learn.
> - Moreover, we have mentioned that there are some other methods utilizing Fourier for domain generalization in Section 2.3 of the main paper. However, none of them directly utilizes the Fourier phase to guide the part of feature learning (internally invariant features).
> - Empirical results comparison with FACT:
>
> We also re-implement FACT for fair comparisons. The results can be found in
> the following table. Note that there are some differences, e.g. data splits, optimizers, and some other tricks, between the official implementation and ours. Our implementation cannot achieve the same results mentioned in the original paper, but we can ensure that comparisons are fair. We tune the hyperparameter for all methods and some implementations can be found in [DeepDG](https://github.com/jindongwang/transferlearning/tree/master/code/DeepDG). From the table, we can see that ours has better performance, especially for time-series benchmarks which are ignored by most methods.
>
> | Digits-DG      | M     | MM      | SV      | SY    | AVG       | DSADS     | 0     | 1     | 2     | 3     | AVG       |
> |----------------|-------|---------|---------|-------|-----------|-----------|-------|-------|-------|-------|-----------|
> | FACT           | 97.65 | 60.42   | 59.97   | 89.52 | 76.89     | FACT      | 84.74 | 88.38 | 90.61 | 78.42 | 85.54     |
> | Ours           | 97.82 | 57.9    | 64.3    | 89.98 | **77.5**  | Ours      | 94.3  | 87.24 | 92.15 | 85.35 | **89.76** |
> | PACS           | A     | C       | P       | S     | AVG       | USC       | 0     | 1     | 2     | 3     | AVG       |
> | FACT           | 77.54 | 76.62   | 88.32   | 81.85 | 81.08     | FACT      | 79.49 | 62.58 | 77.58 | 61.73 | 70.35     |
> | Ours           | 80.86 | 77.6    | 95.57   | 79.49 | **83.38** | Ours      | 83.52 | 69.16 | 76.45 | 79.42 | **77.14** |
> | VLCS           | C     | L       | S       | V     | AVG       | PAMAP     | 0     | 1     | 2     | 3     | AVG       |
> | FACT           | 92.93 | 63.86   | 67.09   | 64.9  | 72.19     | FACT      | 88.09 | 82.55 | 43.71 | 83.86 | 74.55     |
> | Ours           | 96.61 | 67.21   | 74.31   | 75.24 | **78.34** | Ours      | 90.65 | 82.35 | 59.9  | 89.25 | **80.54** |
> | Cross-Dataset  | DSADS | USC-HAD | UCI-HAR | PAMAP | AVG       | EMG       | 0     | 1     | 2     | 3     | AVG       |
> | FACT           | 31.28 | 45.07   | 42.75   | 15.88 | 33.75     | FACT      | 50.06 | 53.91 | 57.66 | 48.91 | 52.63     |
> | Ours           | 46.9  | 49.28   | 46.44   | 52.72 | **48.83** | Ours      | 71.83 | 82.8  | 76.91 | 75.84 | **76.84** |
> | Cross-Position | 0     | 1       | 2       | 3     | 4         | AVG       |       |       |       |       |           |
> | FACT           | 45.89 | 22.37   | 26.85   | 19.01 | 13.27     | 25.48     |       |       |       |       |           |
> | Ours           | 49.55 | 32.73   | 41.75   | 33.43 | 34.2      | **38.33** |       |       |       |       |           |

---

> > ### Comment · Reviewer_B7sJ · 2022-07-01
> > **Thank you for the answers**
> >
> > I thank the authors for their replies, answering most of my initial doubts. However, I have still some remaining concerns:
> > 1. I understand the concerns of the authors. However, for fairness, I would still/also report the results of the original approaches taken from their published papers (when the same split/setup is used). This ensures that the results reported in this work create inconsistencies with the existing literature (i.e. clarifying to the reader the existing discrepancies).
> > 2. The caption of Fig.7 should explicitly mention which experiment it refers to. It would also be helpful to show the average results (or results on multiple setups, even multi-domain DG) in Fig. 7 to confirm that the findings of the ablation are consistent across scenarios.

---

> > > ### Author Response · Authors · 2022-07-04
> > > **Further Response to Reviewer B7sJ**
> > >
> > > We are glad that you acknowledge our response. To further alleviate your concerns, we report more results here.
> > >
> > > Precisely speaking, we fully understand that fair comparisons are extremely important. Thus, we tried our best to implement our method on their benchmarks to ensure fairness.
> > >
> > > 1. Compare our method to FACT.
> > >
> > > We reimplement our method on PACS to compare with the original results in the paper. The results are in the following. As we can see, our method still achieves the best results. Please note that few papers pay attention to time-series benchmarks in DG.
> > >
> > > | Methods           | A          | C          | P          | S          | AVG        |
> > > |-------------------|------------|------------|------------|------------|------------|
> > > | DeepAll           | 77.63      | 76.77      | 95.85      | 69.50      | 79.94      |
> > > | L2A-OT            | 83.30      | 78.20      | 96.20      | 73.60      | 82.80      |
> > > | RSC               | 83.43      | **80.31** | 95.99      | **80.85** | 85.15      |
> > > | RSC(their imple.) | 80.55      | 78.60      | 94.43      | 76.02      | 82.40      |
> > > | FACT              | 85.37      | 78.38      | 95.15      | 79.15      | 84.51      |
> > > | Ours              | **85.76** | 78.92      | **96.71** | 80.15      | **85.39** |
> > >
> > > 2. More results in multi-domain DG.
> > >
> > > We have revised the caption of Fig.7 in the revised version. In the original paper, we have shown some results of ablation study in multi-domain DG, e.g. Fig. 6 and Tab. 9. Here, we report more results of parameter sensitivity on PACS in multi-domain DG. Please note that we have reported the average results for ablation study. For parameter sensitivity, it might be better to report the results for specific tasks since different tasks may have different hyperparameters. As we can see in the following table, our method achieves better performance near the optimal point, demonstrating that our method is insensitive to some hyperparameter choices to some extent.
> > >
> > > | Value   | 0.001 | 0.01      | 0.1       | 0.5   | 1     | 10    |
> > > |---------|-------|-----------|-----------|-------|-------|-------|
> > > | ERM     | 77.86 | 77.86     | 77.86     | 77.86 | 77.86 | 77.86 |
> > > | lambda1 | 78.47 | **79.49** | 78.16     | 77.41 | 74.5  | 66.43 |
> > > | lambda2 | -     | **79.49** | 78.49     | 78.54 | 76.23 | 75.67 |
> > > | lambda3 | -     | 78.07     | **79.49** | 77.9  | 73.72 | -     |

---

> > > > ### Comment · Reviewer_B7sJ · 2022-07-05
> > > > **Thank you and details on the concern**
> > > >
> > > > I thank the authors for reporting additional results in the response.  In my perspective (for the reasons described above), I would consider the article ready for publication if:
> > > > 1. The expanded ablation on \lambda values is included in the manuscript (even in the appendix).
> > > > 2. Table 1-3 report (for each method for which this is available) the results of the original published paper for the same architecture/setup (as done for RSC in the example above).
> > > > 3. Minor: the caption of Fig. 7 describes which setting (i.e. source/target) is used for the ablation (at the moment it is not explicitly stated).

---

> > > > > ### Author Response · Authors · 2022-07-05
> > > > > **Further Response to Reviewer B7sJ**
> > > > >
> > > > > Thank you for your acknowledgment. We have carefully revised our paper.

---

> > > > > > ### Comment · Reviewer_B7sJ · 2022-07-05
> > > > > > **Thanks! final suggestion**
> > > > > >
> > > > > > Thanks! I apologize for not noticing before, but I have another small suggestion: at the moment, the comparison with FACT is only in Section A.1 of the appendix. Given that this is the main comparison with Fourier approaches, it should be included in the paper as well (i.e. the results in "Response for Reviewer Mw6s (3)", including, in parentheses the results of the original paper).

---

> > > > > > > ### Author Response · Authors · 2022-07-05
> > > > > > > **Further Response to Reviewer B7sJ**
> > > > > > >
> > > > > > > Thank you for your suggestion. We have carefully revised our paper.

---

> ### Author Response · Authors · 2022-06-11
> **Response for Reviewer B7sJ (2)**
>
> 3. The weight of feature exploration in Fig. 7c
>
> In fact, we have some hints of the trend in Ablation study. W./o Exp means that $\lambda_3=0$. In this situation, the results are slightly worse than the results with $\lambda_3>0$. To alleviate your concerns, we add some more results.
>
> | lambda3 | -0.01 | 0     | 0.001 | 0.01  | 0.1   | 0.5  | 1    |
> |---------|-------|-------|-------|-------|-------|------|------|
> | Ours    | 82.63 | 85.75 | 85.92 | 86.17 | 81.98 | 77.9 | 78.2 |
>
> Moreover, note that it achieves better performances when $\lambda_3$ is small. It is the limitation of the selection of d in Eq.6. We have tried some other d in Appendix. And it is our future research direction.
>
> 4. Fair comparison and implementation details.
>
> We try our best to ensure fair comparisons. As mentioned in the main paper, we utilize the same backbone, the same optimizer, the same splits, and some other same tricks, e.g. data transformation (The epoch is so large that each method can reach convergence.). We tune hyperparameters for each method and select the best to compare (we even tune some base hyperparameters, e.g. lr, for some methods but we report the mainstream of the base hyperparameters.). Some implementations can be found in [DeepDG](https://github.com/jindongwang/transferlearning/tree/master/code/DeepDG).
>
> Moreover, note that DomainBed is a popular benchmark but not the only one. Domainbed utilizes Adam and totally random parameters, which are not suitable for some methods. Therefore, lots of methods, e.g. FACT, do not utilize it. In addition, utilizing the results reported in the original paper are also not suitable. Different tricks, different data splits, and even different initial states can have large influences on the final results (As shown in DomainBed, in the totally same and random environment, some methods decline seriously.). We try to get a balance between totally random and good performance. We believe ours can achieve better performances with more careful tuning. But we focus on methods and fairness.
>
> 5. Why the numbers are higher than what is reported in the tables in Fig.7.
>
> We do not report the average performance but the performance of one task in Fig.7. For better performance, we tune hyperparameters for each method in each task but not in the whole benchmark. And Fig.7 corresponds to the third task in Table 8.
>
> 6. The meaning of internal and mutual invariance.
>
> Internal invariance means that the features can be learned with a single domain and the features contain intrinsic semantics which are mainly for classification, i.e., the property within a domain, which is agnostic to other domains.
> Mutually invariance means that the features can be learned with multiple domains (cross-domain) and the features contain common information, i.e., the transferable features w.r.t. other domains.
>
> #### Requested Changes.
>
> We will revise the paper accordingly.
>
> 1. Improving the clarity of the manuscript for what concerns the definitions (point 6 above) and methodological components (point 1 above).
>
> Thank you for your advice. We have answered it and we will revise our paper.
>
> 2. Expand the comparison with related works (point 2 above) and provide the performance of other approaches when hyperparameters are tuned for their specific design choices, at least when available already in the literature (see point 4 above).
>
> Thank you for your advice. We have answered it and we will revise our paper.
>
> 3. Provide more details/clarifications (point 5) and expand the ablation analysis (point 3) on Fig.7.
>
> Thank you for your advice. We have answered it and we will revise our paper.

---

### Review · Reviewer_Mw6s · 2022-06-16

**Summary Of Contributions:**

This paper is built upon the assumption that domain-invariant features should be originating both internally and mutually. The internal representation captures the intrinsic/private semantics of the data, and the mutual ones conclude the domain-invariant knowledge. The authors propose a framework DIFEX with the help of knowledge distillation to learn the high-level Fourier phase as internal features and the cross-domain alignment for domain-invariant learning. The proposed model has been evaluated over multiple benchmarks, including Digits, PACS, and VLCS, with competitive performance.

**Broader Impact Concerns:**

I have not seen any concerns about border impact.

**Requested Changes:**

Suggestions:

1. Please consider including more recent methods for comparison.

2. And some critical benchmarks, such as Office-home or DomainNet, are missing.

**Strengths And Weaknesses:**

Strengths:

+ Knowledge distillation has not been widely applied to the DG problem. Using the Fourier transformation to extract the high-frequent features seems a straightforward way for internal representation extraction.

+ The overall framework is simple and effective. It has outperformed most of the baseline methods by a considerable margin. There are solid experiments, including multi/single domain DG and EMG tasks with detailed analyses to enrich the content of this paper.

+ This paper is written in clear logic with a comprehensive survey of related literature. Even the most recent works are included in the discussion.

Weaknesses:

- From my point of view, separating the raw representations into domain private/shared ones is not new. Similar ideas have been well explored in domain adaptation/few-shot learning communities. (I have not listed the refs here, but a similar concept is pretty popular.)

- Applying the high-level Fourier features is also not new in the DG area [1, 2]. Despite the differences claimed in the paper, the overall contributions/novelties are marginal.

[1] Test-time Fourier Style Calibration for Domain Generalization. IJCAI 22.

[2] A Fourier-based Framework for Domain Generalization. CVPR 21.

- In some DG scenarios, the proposed DIFEX cannot outperform the previous methods. Some recent techniques, such as [1,2], are suggested to be included for comparison in PACS. And some critical benchmarks, such as Office-home or DomainNet, are missing.

---

> ### Author Response · Authors · 2022-06-21
> **Response for Reviewer Mw6s (1)**
>
> Thank you for your professional and constructive feedback. We are very glad that you acknowledge our idea, writing, analysis, and comprehensive experiments.
>
> 1. "Separating the raw representations into domain private/shared ones is not novel."
>
> Indeed, learning the private and shared representations, or disentanglement, is a quite popular research direction for domain adaptation/generalization. However, the novelty is never the disentanglement idea itself, but **how we can achieve such representation separation**. From this point of view, our method is novel since it is the first work that splits the representations into *internally* and *mutually* invariant aspects. Most of the existing disentanglement works are based on a variational autoencoder structure, which is totally different from ours.
>
> Therefore, we would like you to pay attention to the specific methodology instead of the general idea.
>
> 2. The novelty of applying the high-level Fourier features.
>
> Yes, we are not the first work that adopts Fourier features (we also did not claim that). Fourier features are also popular in recent years, but the problem is that the novelty lies in **how to adopt Fourier features for DG**. To this end, our approach is totally different from existing research:
>
> First, please note that [2] is not for a traditional DG setting. The model changes when testing. [3] utilized Fourier amplitude for data augmentation. Our method **directly utilizes knowledge distillation** to extract internally-invariant features, i.e. Fourier phase information. It is no doubt that ours has a **totally different implementation**.
>
> Second, applying the high-level Fourier features is only part of our method. It is for internally-invariant features. Our method also includes mutually-invariant feature learning. These **concepts** and the **combination** are novel. Moreover, we also include **exploration**. To the best of our knowledge, few methods pay attention to the exploration among different types of features for diversity.
>
> [2] Test-time Fourier Style Calibration for Domain Generalization. IJCAI 22.
>
> [3] A Fourier-based Framework for Domain Generalization. CVPR 21.

---

> > ### Comment · Reviewer_Mw6s · 2022-07-12
> > **Post Rebuttal**
> >
> > Dear Authors,
> >
> > Thanks for your detailed rebuttal. Most of my concerns are addressed in the response. Please make sure to include the latest experiments and discussion of the contributions in the new manuscript.
> >
> > Thanks

---

> ### Author Response · Authors · 2022-06-21
> **Response for Reviewer Mw6s (2)**
>
> 3. More comparison methods and benchmarks.
>
> First, please note that our proposed DIFED outperforms all the methods mentioned in the main paper with our implementation. We implement all methods for **complete fairness**. Since different data splits, different environments, different optimizers, and some other factors all have influences on the final performance, the results reported in the original papers do not mean anything (e.g. in DomainBed [4], ERM even performs better than some of latest methods.).
>
> Second, it is not fair to compare with [2] since [2] *changes* the model when testing. We have compared our method to many latest methods. To further alleviate your concerns, we re-implement FACT [3] and perform a fair comparison with it. Our method still achieves the best average performance.
>
> | Digits-DG      | M     | MM      | SV      | SY    | AVG       | DSADS     | 0     | 1     | 2     | 3     | AVG       |
> |----------------|-------|---------|---------|-------|-----------|-----------|-------|-------|-------|-------|-----------|
> | FACT           | 97.65 | 60.42   | 59.97   | 89.52 | 76.89     | FACT      | 84.74 | 88.38 | 90.61 | 78.42 | 85.54     |
> | Ours           | 97.82 | 57.9    | 64.3    | 89.98 | **77.5**  | Ours      | 94.3  | 87.24 | 92.15 | 85.35 | **89.76** |
> | PACS           | A     | C       | P       | S     | AVG       | USC       | 0     | 1     | 2     | 3     | AVG       |
> | FACT           | 77.54 | 76.62   | 88.32   | 81.85 | 81.08     | FACT      | 79.49 | 62.58 | 77.58 | 61.73 | 70.35     |
> | Ours           | 80.86 | 77.6    | 95.57   | 79.49 | **83.38** | Ours      | 83.52 | 69.16 | 76.45 | 79.42 | **77.14** |
> | VLCS           | C     | L       | S       | V     | AVG       | PAMAP     | 0     | 1     | 2     | 3     | AVG       |
> | FACT           | 92.93 | 63.86   | 67.09   | 64.9  | 72.19     | FACT      | 88.09 | 82.55 | 43.71 | 83.86 | 74.55     |
> | Ours           | 96.61 | 67.21   | 74.31   | 75.24 | **78.34** | Ours      | 90.65 | 82.35 | 59.9  | 89.25 | **80.54** |
> | Cross-Dataset  | DSADS | USC-HAD | UCI-HAR | PAMAP | AVG       | EMG       | 0     | 1     | 2     | 3     | AVG       |
> | FACT           | 31.28 | 45.07   | 42.75   | 15.88 | 33.75     | FACT      | 50.06 | 53.91 | 57.66 | 48.91 | 52.63     |
> | Ours           | 46.9  | 49.28   | 46.44   | 52.72 | **48.83** | Ours      | 71.83 | 82.8  | 76.91 | 75.84 | **76.84** |
> | Cross-Position | 0     | 1       | 2       | 3     | 4         | AVG       |       |       |       |       |           |
> | FACT           | 45.89 | 22.37   | 26.85   | 19.01 | 13.27     | 25.48     |       |       |       |       |           |
> | Ours           | 49.55 | 32.73   | 41.75   | 33.43 | 34.2      | **38.33** |       |       |       |       |           |
>
> Third, we think we have included lots of popular critical benchmarks that can prove the superiority of our method. We evaluate **8** datasets (**three popular visual datasets** and five popular time-series datasets) in **two modalities** (visual, time-series) with **two different settings** (single, multiple). Comprehensive experiments have demonstrated the superiority of our method. To further alleviate your concerns, we also show some results on OfficeHome here. Our method still achieves the best average performance.
>
> | OfficeHome | A          | C          | P          | R          | AVG        |
> |------------|------------|------------|------------|------------|------------|
> | ERM        | 58.63      | 47.95      | 72.22      | 73.03      | 62.96      |
> | DANN       | 57.73      | 44.42      | 71.95      | 72.50      | 61.65      |
> | CORAL      | 58.76      | 48.75      | 72.34      | 73.63      | 63.37      |
> | Mixup      | 55.79      | 47.88      | 71.95      | 72.83      | 62.11      |
> | GroupDRO   | 57.60      | 48.77      | 71.53      | 73.17      | 62.77      |
> | RSC        | 58.96      | 49.16      | 72.54      | **74.16** | 63.70      |
> | ANDMask    | 56.74      | 45.86      | 70.67      | 73.19      | 61.61      |
> | Ours       | **59.79** | **49.93** | **73.46** | 74.00      | **64.30** |
>
> [4] Gulrajani, Ishaan, and David Lopez-Paz. "In Search of Lost Domain Generalization." International Conference on Learning Representations. 2021.

---

> ### Author Response · Authors · 2022-06-21
> **Response for Reviewer Mw6s (3)**
>
> #### Requested Changes.
>
> We will revise the paper accordingly.
>
> 1. Consider including more recent methods for comparison.
>
> To further alleviate your concerns, we also compare our method to Mixstyle [5] and SWAD [6]. Our method still achieves the best average performance.
>
> | Digits-DG      | M     | MM      | SV      | SY    | AVG       | DSADS     | 0     | 1     | 2     | 3     | AVG       |
> |----------------|-------|---------|---------|-------|-----------|-----------|-------|-------|-------|-------|-----------|
> | Mixstyle       | 96.58 | 60.32   | 55.87   | 87.92 | 75.17     | Mixstyle  | 90.18 | 78.29 | 87.76 | 75.13 | 82.84     |
> | SWAD+ERM       | 97.37 | 55.77   | 59      | 87.92 | 75.02     | SWAD+ERM  | 89.69 | 76.8  | 86.93 | 80.13 | 83.39     |
> | Ours           | 97.82 | 57.9    | 64.3    | 89.98 | **77.5**  | Ours      | 94.3  | 87.24 | 92.15 | 85.35 | **89.76** |
> | PACS           | A     | C       | P       | S     | AVG       | USC       | 0     | 1     | 2     | 3     | AVG       |
> | Mixstyle       | 83.11 | 78.37   | 93.65   | 76.23 | 82.84     | Mixstyle  | 75.62 | 53.6  | 78.21 | 57.55 | 66.25     |
> | SWAD+ERM       | 81.93 | 76.45   | 94.67   | 76.79 | 82.46     | SWAD+ERM  | 82.4  | 56.56 | 71.07 | 69.08 | 69.78     |
> | Ours           | 80.86 | 77.6    | 95.57   | 79.49 | **83.38** | Ours      | 83.52 | 69.16 | 76.45 | 79.42 | **77.14** |
> | VLCS           | C     | L       | S       | V     | AVG       | PAMAP     | 0     | 1     | 2     | 3     | AVG       |
> | Mixstyle       | 95.83 | 65.02   | 68.25   | 72.33 | 75.36     | Mixstyle  | 82.89 | 76.15 | 53.48 | 79.21 | 72.93     |
> | SWAD+ERM       | 94.56 | 61.78   | 67.82   | 67.89 | 73.01     | SWAD+ERM  | 85.18 | 74.93 | 54.06 | 81.78 | 73.99     |
> | Ours           | 96.61 | 67.21   | 74.31   | 75.24 | **78.34** | Ours      | 90.65 | 82.35 | 59.9  | 89.25 | **80.54** |
> | Cross-Dataset  | DSADS | USC-HAD | UCI-HAR | PAMAP | AVG       | EMG       | 0     | 1     | 2     | 3     | AVG       |
> | Mixstyle       | 7.18  | 39.37   | 37.96   | 36.04 | 30.14     | Mixstyle  | 64.26 | 71.72 | 68.48 | 70.53 | 68.75     |
> | SWAD+ERM       | 28.3  | 40.99   | 43.94   | 16.11 | 32.33     | SWAD+ERM  | 42.02 | 57.06 | 53.89 | 48.73 | 50.42     |
> | Ours           | 46.9  | 49.28   | 46.44   | 52.72 | **48.83** | Ours      | 71.83 | 82.8  | 76.91 | 75.84 | **76.84** |
> | Cross-Position | 0     | 1       | 2       | 3     | 4         | AVG       |       |       |       |       |           |
> | Mixstyle       | 36.32 | 25.13   | 35.27   | 25.93 | 19.73     | 28.48     |       |       |       |       |           |
> | SWAD+ERM       | 43.83 | 30.18   | 39.35   | 25.38 | 27.58     | 33.26     |       |       |       |       |           |
> | Ours           | 49.55 | 32.73   | 41.75   | 33.43 | 34.2      | **38.33** |       |       |       |       |           |
>
> [5] Zhou, Kaiyang, Yongxin Yang, Yu Qiao, and Tao Xiang. "Domain generalization with mixstyle." ICLR (2021).
>
> [6] Cha, Junbum, Sanghyuk Chun, Kyungjae Lee, Han-Cheol Cho, Seunghyun Park, Yunsung Lee, and Sungrae Park. "Swad: Domain generalization by seeking flat minima." Advances in Neural Information Processing Systems 34 (2021).
>
> 2. Some critical benchmarks, such as Office-home or DomainNet, are missing.
>
> Thank you for your advice. We have answered it.

---

### Author Response · Authors · 2022-06-21
**New revision**

Dear ACs and reviewers,

Thank you for your constructive feedback! Now we upload a new revision of the paper according to your comments, which contains the following modifications:
1. We add more explanations and details to better describe our work.
2. We thoroughly re-examined all the minor weaknesses mentioned by reviewers and modified them.
3. We add more comparison methods to demonstrate the superiority of our method.
4. We resolved many questions and concerns raised by reviewers.

We hope that you can like it. If you have any questions, please reply to us:)

---

### Decision · Action_Editors · 2022-07-15

**Recommendation:** Accept as is

**Comment:**

This paper proposes a new domain generalization method that explores internal invariance together with domain invariance. To this end, the proposed method relies on a knowledge distillation framework to capture the high-level Fourier phase as the internally-invariant features and learn cross-domain correlation alignment as the mutually-invariant features.

The reviewers agree that this paper makes novel contributions to domain generalization. Especially, the internal invariance idea has not been explored in previous works. Reviewers has some concerns on the novelty of some components, such as the Fourier method and the domain-shared and specific separation, and the authors have addressed these concerns in the discussion. Also, the authors also added lots of new experimental studies as suggested by the reviewers. Finally, all the reviewers vote for acceptance of this paper.

I would recommend acceptance of this paper given its novel contribution and solid empirical results (after revision).